# Formulating Robustness Against Unforeseen Attacks

**Sihui Dai**
Princeton University
sihuid@princeton.edu

**Saeed Mahloujifar**
Princeton University
sfar@princeton.edu

**Prateek Mittal**
Princeton University
pmittal@princeton.edu

## Abstract

Existing defenses against adversarial examples such as adversarial training typically assume that the adversary will conform to a specific or known threat model, such as $\ell_p$ perturbations within a fixed budget. In this paper, we focus on the scenario where there is a mismatch in the threat model assumed by the defense during training, and the actual capabilities of the adversary at test time. We ask the question: if the learner trains against a specific "source" threat model, when can we expect robustness to generalize to a stronger unknown "target" threat model during test-time? Our key contribution is to formally define the problem of learning and generalization with an unforeseen adversary, which helps us reason about the increase in adversarial risk from the conventional perspective of a known adversary. Applying our framework, we derive a generalization bound which relates the generalization gap between source and target threat models to variation of the feature extractor, which measures the expected maximum difference between extracted features across a given threat model. Based on our generalization bound, we propose variation regularization (VR) which reduces variation of the feature extractor across the source threat model during training. We empirically demonstrate that using VR can lead to improved generalization to unforeseen attacks during test-time, and combining VR with perceptual adversarial training (Laidlaw et al., 2021) achieves state-of-the-art robustness on unforeseen attacks. Our code is publicly available at https://github.com/inspire-group/variation-regularization.

## 1 Introduction

Neural networks have impressive performance on a variety of datasets (LeCun et al., 1998; He et al., 2015; Krizhevsky et al., 2017; Everingham et al., 2010) but can be fooled by imperceptible perturbations known as adversarial examples (Szegedy et al., 2014). The conventional paradigm to mitigate this threat often assumes that the adversary generates these examples using some known threat model, primarily $\ell_p$ balls of specific radius (Cohen et al., 2019; Zhang et al., 2020b; Madry et al., 2018), and evaluates the performance of defenses based on this assumption. This assumption, however, is unrealistic in practice. In general, the learner does not know exactly what perturbations the adversary will apply during test-time.

To bridge the gap between the setting of robustness studied in current adversarial ML research and robustness in practice, we study the problem of *learning with an unforeseen adversary*. In this problem, the learner has access to adversarial examples from a proxy "source" threat model but wants to be robust against a more difficult "target" threat model used by the adversary during test-time. We ask the following questions:

1. *When can we expect robustness on the source threat model to generalize to the true unknown target threat model used by the adversary?*

2. *How can we design a learning algorithm that reduces the drop in robustness from source threat model to target threat model?*

36th Conference on Neural Information Processing Systems (NeurIPS 2022).

To address the first question, we introduce *unforeseen adversarial generalizability* which provides a framework for reasoning about what types of learning algorithms produce models that generalize well to unforeseen attacks. Based on this framework, we derive a generalization bound which relates the difference in adversarial risk across source and target threat models to a quantity we call variation: the expected maximum difference between extracted features across a given threat model.

Our bound addresses the second question; it suggests that learning algorithms that bias towards models with small variation across the source threat model exhibit smaller drop in robustness to particular unforeseen attacks. Thus, we propose variation regularization (VR) to improve robustness to unforeseen attacks. We then empirically demonstrate that when combined with adversarial training, VR improves generalization to unforeseen attacks during test-time across multiple datasets and architectures. Our contributions are as follows:

**We formally define the problem of learning with an unforeseen adversary with respect to adversarial risk.** We make the case that one way of learning with an unforeseen adversary is to ensure that the gap between the empirical adversarial risk measured on the source adversary and the expected adversarial risk on the target adversary remains small. To this end, we define *unforeseen adversarial generalizability* which provides a framework for understanding under what conditions we would expect small generalization gap.

**Under our framework for generalizability, we derive a generalization bound for generalization across threat models.** Our bound relates the generalization gap to a quantity we define as variation, the expected maximum difference between extracted features across a given threat model. We demonstrate that under certain conditions, we can decrease this upper bound *while only using information about the source threat model*.

**Using our bound, we propose a regularization term which we call variation regularization (VR).** We incorporate this regularization term into adversarial training and perceptual adversarial training (Laidlaw et al., 2021), leading to learning algorithm that we call AT-VR and PAT-VR respectively. We find that VR can lead to improved robustness on unforeseen attacks across datasets such as CIFAR-10, CIFAR-100, and ImageNette over adversarial training without VR. Additionally, PAT-VR achieves state-of-the-art (SOTA) robust accuracy on LPIPS-based attacks, improving over PAT by 21% and SOTA robust accuracy on a union of $\ell_\infty$, $\ell_2$, spatially transformed (Xiao et al., 2018), and recolor attacks (Laidlaw & Feizi, 2019).

## 2 Related Works

**Adversarial examples and defenses** Previous studies have shown that neural networks can be fooled by perturbations known as adversarial examples, which are imperceptible to humans but cause NNs to predict incorrectly with high confidence (Szegedy et al., 2014). These adversarial examples can be generated by various threat models including $\ell_p$ perturbations, spatial transformations (Xiao et al., 2018), recoloring (Laidlaw & Feizi, 2019), and broader threat models such as fog and snow distortions (Kang et al., 2019). While many defenses have been proposed, most defenses provide guarantees for specific threat models (Cohen et al., 2019; Zhang et al., 2020b; Croce & Hein, 2020a; Yang et al., 2020; Zhang et al., 2020a) or use knowledge of the threat model during training (Madry et al., 2018; Zhang et al., 2019; Wu et al., 2020a; Tramèr & Boneh, 2019; Maini et al., 2020). Adversarial training is a popular defense framework in which a model is trained using adversarial examples generated by a particular threat model, such as $\ell_2$ or $\ell_\infty$ attacks (Madry et al., 2018; Zhang et al., 2019; Wu et al., 2020a). Prior works have also extended adversarial training to defend against unions of attack types such as unions of $\ell_p$-balls (Maini et al., 2020; Tramèr & Boneh, 2019) and against stronger adversaries more aligned to human perception (Laidlaw et al., 2021).

**Bounds for Learning with Adversarial Examples** An interesting body of work studies generalization bounds for specific attacks Cullina et al. (2018); Attias et al. (2019); Montasser et al. (2019); Raghunathan et al. (2019); Chen et al. (2020); Diakonikolas et al. (2019); Yu et al. (2021); Diochnos et al. (2019). In particular, they study generalization in the setting where the learning algorithm minimizes the adversarial risk on the training set and hopes to generalize to same adversary during test-time. Montasser et al. (2021) provide bounds for the problem of generalizing to an unknown adversary with oracle access to that adversary during training. Our work differs since we study generalization and provide bounds under the setting in the learner only has access to samples from a weaker adversary than present at test-time.

**"Unforeseen" attacks and defenses** While several prior works have studied "unforeseen" attacks (Kang et al., 2019; Stutz et al., 2020; Laidlaw et al., 2021; Jin & Rinard, 2020), these works are empirical works, and the term "unforeseen attack" has not been formally defined. Kang et al. (2019) first used the term "unforeseen attack" when proposing a set of adversarial threat models including Snow, Fog, Gabor, and JPEG to evaluate how well defenses can generalize from $\ell_\infty$ and $\ell_2$ to broader threat models. Stutz et al. (2020) and Chen et al. (2022) propose adversarial training based techniques with a mechanism for abstaining on certain inputs to improve generalization from training on $\ell_\infty$ to *stronger* attacks including $\ell_p$ attacks of larger norm. Other defenses against "unforeseen attacks" consider them to be attacks that are not used during training, but not necessarily stronger than those used in training. For instance, Laidlaw et al. (2021) propose using LPIPS (Zhang et al., 2018), a perceptually aligned image distance metric, to generate adversarial examples during training. They demonstrate that by training using adversarial examples using this distance metric, they can achieve robustness against a variety of adversarial threat models including $\ell_\infty$, $\ell_2$, recoloring, and spatial transformation. However, the LPIPS attack is the strongest out of all threat models tested and contains a large portion of those threat models. To resolve these differences in interpretation of "unforeseen attack", we provide a formal definition of learning with an unforeseen adversary.

**Domain Generalization** A problem related to generalizing to unforeseen attacks is the problem of domain generalization under covariate shift. In the domain generalization problem, the learner has access to multiple training distributions and has the goal of generalizing to an unknown test distribution. (Albuquerque et al., 2019) demonstrate that when the test distribution lies within a convex hull of the training distributions, learning is feasible. (Ye et al., 2021) propose a theoretical framework for domain generalization in which they derive a generalization bound in terms of the variation of features across training distributions. We focus on the problem of generalizing to unforeseen adversaries and demonstrate that a generalization bound in terms of variation of features across the training threat model exists.

# 3 Adversarial Learning with an Unforeseen Adversary

**Notations** We use $\mathcal{D} = (\mathcal{X}, \mathcal{Y})$ to denote the data distribution and $D_m$ to denote a dataset formed by $m$ iid samples from $\mathcal{D}$. We use $X$ to denote the support of $\mathcal{X}$. To match learning theory literature, we will refer to the defense as a learning algorithm $\mathcal{A}$, which takes the adversarial threat model and training data as inputs and outputs the learned classifier ($\hat{f} = \mathcal{A}(S, D_m)$ where $S$ is the threat model). We use $\mathcal{F}$ to denote the function class that $\mathcal{A}$ is applied over and $\mathcal{A}(S, D_m) \in \mathcal{F}$. $\mathcal{F} = \mathcal{G} \circ \mathcal{H}$ denotes a function class where $\forall f \in \mathcal{F}$, $f = g \circ h$ where $g \in \mathcal{G}$, $h \in \mathcal{H}$.

In this section, we will define what constitutes an unforeseen attack and the learner's goal in the presence of unforeseen attacks. We then introduce unforeseen adversarial generalizability which provides a framework for reasoning about what types of learning algorithms give models that generalize well to unforeseen adversaries.

## 3.1 Formulating Learning with an Unforeseen Adversary

To formulate adversarial learning with an unforeseen adversary, we begin by defining threat model and adversarial risk. We will then use these definitions to explain the goal of the learner in the presence of an unforeseen adversary.

**Definition 3.1** (Threat Model). The *threat model* is defined by a neighborhood function $N(\cdot) : X \to \{0,1\}^X$. For any input $x \in X$, $N(x)$ contains $x$.

**Definition 3.2** (Expected and Empirical Adversarial Risk). We define *expected adversarial risk* for a model $f$ with respect to a threat model $N$ as $L_N(f) = \mathbb{E}_{(x,y)\sim\mathcal{D}} \max_{x'\in N(x)} \ell(f(x'), y)$ where $\ell$ is a loss function. In practice, we generally do not have access to the true data distribution $\mathcal{D}$, but have $m$ iid samples $\{(x_i, y_i)\}_{i=1}^m$. We can approximate $L_N(f)$ with the *empirical adversarial risk* defined as $\hat{L}_N(f) = \frac{1}{m}\sum_{i=1}^m \max_{x_i'\in N(x_i)} \ell(f(x_i'), y_i)$

In adversarial learning, the learner's goal is to find a function $f \in \mathcal{F}$ that minimizes $L_T$ where $T$ threat model used by the adversary. We call $T$ the *target threat model*. We call the threat model that the learner has access to during training the *source threat model*. We divide the adversarial learning problem into 2 cases, learning with a foreseen adversary and learning with an unforeseen adversary. To distinguish between these 2 cases, we first define the subset operation for threat models.

**Definition 3.3** (Threat Model Subset and Superset). We call a threat model $S$ a subset of another threat model $T$ (and $T$ a superset of $S$) if $S(x) \subseteq T(x)$ almost everywhere in $\mathcal{X}$. We denote this as $S \subseteq T$ (or $T \supseteq S$). If $S(x) \subset T(x)$ almost everywhere in $\mathcal{X}$, then we call $S$ a strict subset of $T$ (and $T$ a strict superset of $S$) and denote this as $S \subset T$ (or $T \supset S$).

**Learning with a Foreseen Adversary** In learning with a foreseen adversary, the target threat model $T$ is a subset of the source threat model $S$ ($T \subseteq S$). The learner has access to $S$ and a dataset $D_m$ of $m$ iid samples from the data distribution $\mathcal{D}$. The learner would like to use a learning algorithm $\mathcal{A}$ for which $f = \mathcal{A}(S, D_m)$ achieves $L_T(f) < \epsilon$ for some small $\epsilon > 0$. The learner cannot compute $L_T(f)$, but can compute $\hat{L}_S(f) \geq \hat{L}_T(f)$. This setting of learning with a foreseen adversary represents when the adversary is weaker than assumed by the learner and since $L_S(f) \geq L_T(f)$, which means that as long as the learner can achieve $L_S(f) < \epsilon$, then they are guaranteed that $L_T(f) < \epsilon$.

**Learning with an Unforeseen Adversary** In learning with an unforeseen adversary, the target threat model $T$ is a strict superset of the source threat model $S$ ($T \supset S$). In this setting, we call $T$ an unforeseen adversary. The learner has access to $S$ and a dataset $D_m$ of $m$ iid samples from the data distribution $\mathcal{D}$. The learner would like to use a learning algorithm $\mathcal{A}$ for which $f = \mathcal{A}(S, D_m)$ achieves $L_T(f) < \epsilon$ for some small $\epsilon > 0$. This setting of learning with an unforeseen adversary represents when the adversary is strictly stronger than assumed by the learner. Compared to learning with a foreseen adversary, this problem is more difficult since $L_S(f)$ may not be reflective of $L_T(f)$. By construction $L_T(f) \geq L_S(f)$, but it is unclear how much larger $L_T(f)$ is. When can we guarantee that $L_T(f)$ is close to $L_S(f)$? We will address this question in the Section 3.2 when we define threat model generalizability and Section 4 when we provide a bound for $L_T(f) - L_S(f)$.

### 3.2 Formulating Generalizability with an Unforeseen Adversary

How should we define $\mathcal{A}$ that performs well against an unforeseen adversary? One way is to have $f = \mathcal{A}(S, D_m)$ achieves small $\hat{L}_S(f)$ (which can be measured by $\mathcal{A}$) while ensuring that $\hat{L}_S(f)$ is close to $L_T(f)$. This leads us to the following definition for generalization gap.

**Definition 3.4** (Generalization Gap). For threat models $S$ and $T$, the generalization gap is defined as $L_T(f) - \hat{L}_S(f)$. We observe that

$$L_T(f) - \hat{L}_S(f) = \underbrace{L_T(f) - L_S(f)}_{\text{threat model generalization gap}} + \underbrace{L_S(f) - \hat{L}_S(f)}_{\text{sample generalization gap}}$$

We note that in the special case of learning with a foreseen adversary, $L_T(f) - L_S(f) \leq 0$, so $L_T(f) - \hat{L}_S(f) \leq L_S(f) - \hat{L}_S(f)$ and bounding the generalization gap be achieved by bounding the sample generalization gap, which has been studied by prior works (Attias et al., 2019; Raghunathan et al., 2019; Chen et al., 2020; Yu et al., 2021).

We would like to ensure that the generalization gap is small with high probability. We can achieve this by ensuring that both the sample generalization gap and threat model generalization gap are small. This leads us to define *robust sample generalizability* and *threat model generalizability* which describe conditions necessary for us to expect the respective generalization gaps to be small. We then combine these generalizability definitions and define *unforeseen adversarial generalizability* which describes the conditions necessary for a learning algorithm to be able to generalize to unforeseen attacks.

**Definition 3.5** (Robust Sample Generalizability). A learning algorithm $\mathcal{A}$ robustly $(\epsilon(\cdot), \delta)$-sample generalizes across function class $\mathcal{F}$ on threat model $S$ where $\epsilon : \mathbb{N} \to \mathbb{R}^+$, if for any distribution $\mathcal{D}$ when running $\mathcal{A}$ on $m$ iid samples $D_m$ from $\mathcal{D}$, we have

$$\mathbb{P}[L_S(\mathcal{A}(S, D_m)) \leq \hat{L}_S(\mathcal{A}(S, D_m)) + \epsilon(m)] \geq 1 - \delta$$

Definition 3.5 implies that any learning algorithm that $(\epsilon(\cdot), \delta)$-robustly sample generalizes across our chosen hypothesis class $\mathcal{F}$ with $\epsilon(m) << 1, \delta << 1$, we can achieve small sample generalization gap with high probability.

We now define generalizability for the threat model generalization gap.

**Definition 3.6** (Threat Model Generalizability). Let $S$ be the source threat model used by the learner. A learning algorithm $\mathcal{A}$ $(\epsilon(\cdot, \cdot), \delta)$-robustly generalizes to target threat model $T$ where

$\epsilon : T \times \mathbb{N} \to \mathbb{R}^+ \cup \{\infty\}$ and $\delta \in [0, 1]$ if for any data distribution $\mathcal{D}$ and any training dataset $D_m$ with $m$ iid samples from $\mathcal{D}$, we have:

$$\mathbb{P}[L_T(\mathcal{A}(S, D_m)) \leq L_S(\mathcal{A}(S, D_m)) + \epsilon(T, m)] \geq 1 - \delta$$

We note that the Definition 3.6 considers generalization to a given $T$, which does not fully account for the unknown nature of $T$, since from the learner's perspective, the learner does not know which threat model it wants $L_T$ to be small for. We address this in the following definition where we combine Definitions 3.5 and 3.6 and define generalizability to unforeseen adversarial attacks.

**Definition 3.7** (Unforeseen Adversarial Generalizability). A learning algorithm $\mathcal{A}$ on function class $\mathcal{F}$ with source adversary $S$, $(\epsilon(\cdot, \cdot), \delta)$-robustly generalizes to unforeseen threat models where $\epsilon : N \times \mathbb{N} \to \mathbb{R}^+ \cup \{\infty\}$ if there exists $\epsilon_1, \epsilon_2$ with $\epsilon_1(m) + \epsilon_2(T, m) \leq \epsilon(T, m)$ such that $\mathcal{A}$ robustly $(\epsilon_1, \delta)$-sample generalizes and $(\epsilon_2, \delta)$-robustly generalizes to *any* threat model $T$.

We remark that in Definition 3.7, $\epsilon$ is a function of $T$, which accounts for differences in difficulty of possible target threat models. Ideally, we would like $\epsilon(T, m)$ at sufficiently large $m$ to be small across a set of reasonable threat models $T$ (ie. imperceptible perturbations) and expect it to be large (and possibly infinite) for difficult or unreasonable threat models (ie. unbounded perturbations).

# 4 A Generalization Bound for Unforeseen Attacks

While prior works have proposed bounds on sample generalization gap (Attias et al., 2019; Raghunathan et al., 2019; Chen et al., 2020; Yu et al., 2021), to the best of our knowledge, prior works have not provided bounds on threat model generalization gap. In this section, we demonstrate that we can bound the threat model generalization gap in terms of a quantity we define as variation, the expected maximum difference across features learned by the model across the target threat model. We then show that with the existence of an expansion function, which relates source variation to target variation, any learning algorithm which with high probability outputs a model with small source variation can achieve small threat model generalization gap.

## 4.1 Relating generalization gap to variation

We now consider function classes of the form $\mathcal{F} = \mathcal{G} \circ \mathcal{H}$ where $\forall g \in \mathcal{G}, g : \mathbb{R}^d \to \mathbb{R}^K$ is a top level classifier into $K$ classes and $\forall h \in \mathcal{H}, h : \mathcal{X} \to \mathbb{R}^d$ is a $d$-dimensional feature extractor. Since the top classifier $g$ is fixed for a function $f$, if $h(\hat{x})_i, i \in [1...d]$ fluctuates a lot across the threat model $\hat{x} \in T(x)$, then the adversary can manipulate this feature to cause misclassification. The relation between features and robustness has been analyzed by prior works such as (Ilyas et al., 2019; Tsipras et al., 2019; Tramèr & Boneh, 2019). We now demonstrate that we can bound the threat model generalization gap in terms of a measure of the fluctuation of $h$ across $T$, which we call *variation*.

**Definition 4.1** (Variation). The variation of a feature vector $h(\cdot) : \mathcal{X} \to \mathbb{R}^d$ across a threat model $N$ is given by

$$\mathcal{V}(h, N) = \mathbb{E}_{(x,y) \sim \mathcal{D}} \max_{x_1, x_2 \in N(x)} ||h(x_1) - h(x_2)||_2$$

**Theorem 4.2** (Variation-Based Threat Model Generalization Bound). *Let $S$ denote the source threat model and $\mathcal{D}$ denote the data distribution. Let $\mathcal{F} = \mathcal{G} \circ \mathcal{H}$ where $\mathcal{G}$ is a class of Lipschitz classifiers with Lipschitz constant upper bounded by $\sigma_{\mathcal{G}}$. Let the loss function be $\rho$-Lipschitz. Consider a learning algorithm $\mathcal{A}$ over $\mathcal{F}$ and denote $f = \mathcal{A}(S, D_m) = g \circ h$. If with probability $1 - \delta$ over the randomness of $D_m$, $\mathcal{V}(h, T) \leq \epsilon(T, m)$ where $\epsilon : T \times \mathbb{N} \to \mathbb{R}^+ \cup \{\infty\}$, then $\mathcal{A}$ $(\rho\sigma_{\mathcal{G}}\epsilon(T, m), \delta)$-robustly generalizes from $S$ to $T$.*

Theorem 4.2 shows we can bound the threat model generalization gap between any source $S$ and unforeseen adversary $T$ in terms of variation across $T$. With regards to Definition 3.6, Theorem 4.2 suggests that any learning algorithm over $\mathcal{F}$ that with high probability outputs models with low variation on the target threat model can generalize well to that target.

## 4.2 Relating source and target variation

Since the learning algorithm $\mathcal{A}$ cannot use information from $T$, it is unclear how to define such $\mathcal{A}$ that achieves small $\mathcal{V}(h, T)$. We address this problem by introducing the notion of an expansion function, which relates the source variation (which can be computed by the learner) to target variation.

**Definition 4.3** (Expansion Function for Variation (Ye et al., 2021)). A function $s : \mathbb{R}^+ \cup \{0\} \rightarrow \mathbb{R}^+ \cup \{0, +\infty\}$ is an expansion function relating variation across source threat model $S$ to target threat model $T$ if the following properties hold:

1. $s(\cdot)$ is monotonically increasing and $s(x) \geq x, \forall x \geq 0$

2. $\lim_{x \to 0^+} s(x) = s(0) = 0$

3. For all $h$ that can be modeled by function class $\mathcal{F}$, $s(\mathcal{V}(h, S)) \geq \mathcal{V}(h, T)$

When an expansion function for variation from $S$ to $T$ exists, then we can bound the threat model generalization gap in terms of variation on $S$. This follows from Theorem 4.2 and Definition 4.3.

**Corollary 4.4** (Source Variation-Based Threat Model Generalization Bound). *Let $S$ denote the source threat model and $\mathcal{D}$ denote the data distribution. Let $\mathcal{F} = \mathcal{G} \circ \mathcal{H}$ where $\mathcal{G}$ is a class of Lipschitz classifiers with Lipschitz constant upper bounded by $\sigma_{\mathcal{G}}$. Let the loss function be $\rho$-Lipschitz. Let $T$ be any unforeseen threat model for which an expansion function $s$ from $S$ to $T$ exists. Consider a learning algorithm $\mathcal{A}$ over $\mathcal{F}$ and denote $f = \mathcal{A}(S, D_m) = g \circ h$. If with probability $1 - \delta$ over the randomness of $D_m$, $s(\mathcal{V}(h, S)) \leq \epsilon(T, m)$ where $\epsilon : T \times \mathbb{N} \rightarrow \mathbb{R}^+ \cup \{\infty\}$, then $\mathcal{A}$ ($\rho \sigma_{\mathcal{G}} \epsilon(T, m), \delta$)-robustly generalizes from $S$ to $T$.*

Corollary 4.4 allows us to relate generalization across threat models of a model $f = g \circ h$ to $s(\mathcal{V}(h, S))$ instead of $\mathcal{V}(h, T)$. While this expression is still dependent on the target threat model $T$ (since $s$ is dependent on $T$), we can reduce $s(\mathcal{V}(h, S))$ *without knowledge of* $T$ due to the monotonicity of the expansion function. Thus, provided that an expansion function exists, we can use techniques such as regularization in order to ensure that our learning algorithm actively chooses models with low source variation. This result leads to the question: when does the expansion function exist?

### 4.3 When does the expansion function exist?

We now demonstrate a few cases in which the expansion function exists or does not exist. We begin by providing basic examples of source threat models $S$ and target threat models $T$ without constraints on function class.

**Proposition 4.5.** *When $S = T$, an expansion function $s$ exists and is given by $s(x) = x$.*

**Proposition 4.6.** *Let $S = \{x\}$, and $T$ be a threat model such that $S \subset T$. Then, for all feature extractors $h$, we have that $V(h, S) = 0$ while $V(h, T)$ can be greater than 0. In this case, no expansion function exists such that $s(V(h, S)) \geq V(h, T)$.*

While we did not consider a constrained function class in the previous two settings, the choice of function class can also impact the existence of an expansion function. For instance, in the setting of Proposition 4.6, if we constrain $\mathcal{F}$ to only use feature extractors with a constant output, then the expansion function $s(x) = x$ is valid. We now consider the case where our function class $\mathcal{F}$ uses linear feature extractors and derive expansion functions for $\ell_p$ adversaries.

**Theorem 4.7** (Linear feature extractors with $\ell_p$ threat model ($p \in \mathbb{N} \cup +\infty$)). *Let inputs $x \in \mathbb{R}^n$ and corresponding label $y \in [1...K]$. Consider $S(x) = \{\hat{x} | ||\hat{x} - x||_p \leq \epsilon_1\}$ and $U(x) = \{\hat{x} | ||\hat{x} - x||_q \leq \epsilon_2\}$ with $p, q \in \mathbb{N}^+, p, q > 0$. Define target threat model $T(x) = S(x) \cup U(x)$. Consider a linear feature extractor with bounded condition number: $h \in \{Wx + b | W \in \mathbb{R}^{d \times n}, b \in \mathbb{R}^d, \frac{\sigma_{\max}(W)}{\sigma_{\min}(W)} \leq B < \infty\}$. Then, an expansion function exists and is linear.*

Theorem 4.7 demonstrates that in the case of a linear feature extractor a linear expansion function exists for any data distribution from a source $\ell_p$ adversary to a union of $\ell_p$ adversaries. This result suggests that with a function class using linear feature extractors, we can improve generalization to $\ell_p$ balls with larger radii by using a learning algorithm that biases towards models with small $\mathcal{V}(h, S)$. We demonstrate this in Appendix D where we experiment with linear models on Gaussian data. We also provide visualizations of expansion function for a nonlinear model (ResNet-18) on CIFAR-10 in Section 5.6.

## 5 Adversarial Training with Variation Regularization

Our generalization bound from Corollary 4.4 suggests that learning algorithms that bias towards small source variation can improve generalization to other threat models when an expansion function exists.

In this section, we propose adversarial training with variation regularization (AT-VR) to improve generalization to unforeseen adversaries and evaluate the performance of AT-VR on multiple datasets and model architectures.

## 5.1 Adversarial training with variation regularization

To integrate variation into AT, we consider the following training objective:

$$\min_{f\in\mathcal{F}, f=g\circ h} \frac{1}{n}\sum_{i=1}^{n}[\underbrace{\max_{x'\in S(x_i)}\ell(f(x'), y_i)}_{\text{empirical adversarial risk}} + \lambda \underbrace{\max_{x', x''\in S(x_i)}||h(x') - h(x'')||_2}_{\text{empirical variation}}]$$

where $\lambda \geq 0$ is the regularization strength. For the majority of our experiments in the main text, we use the objective of PGD-AT (Madry et al., 2018) as the approximate empirical adversarial risk. We note that this can be replaced with other forms of AT such as TRADES (Zhang et al., 2019). We can approximate empirical variation by using gradient-based methods. For example, when $N(x)$ is a $\ell_p$ ball around $x$, we compute the variation term by using PGD to simultaneously optimize over $x_1$ and $x_2$. We discuss methods for computing variation for other source threat models in Appendix E.10.

## 5.2 Experimental Setup

We investigate the performance of training neural networks with AT-VR on image data for a variety of datasets, architectures, source threat models, and target threat models. We also combine VR with perceptual adversarial training (PAT) (Laidlaw et al., 2021), the current state-of-the-art for unforeseen robustness, which uses a source threat model based on LPIPS (Zhang et al., 2018) metric.

**Datasets** We train models on CIFAR-10, CIFAR-100, (Krizhevsky et al., 2009) and ImageNette (Howard). ImageNette is a 10-class subset of ImageNet (Deng et al., 2009).

**Model architecture** On CIFAR-10, we train ResNet-18 (He et al., 2016), WideResNet(WRN)-28-10 (Zagoruyko & Komodakis, 2016), and VGG-16 (Simonyan & Zisserman, 2015) architectures. On ImageNette, we train ResNet-18 (He et al., 2016). For PAT-VR, we use ResNet-50. For all architectures, we consider the feature extractor $h$ to consist of all layers of the NN and the top level classifier $g$ to be the identity function. We include experiments for when we consider $h$ to be composed of all layers before the fully connected layers in Appendix F.

**Source threat models** Across experiments with AT-VR, we consider 2 different source threat models: $\ell_\infty$ perturbations with radius $\frac{8}{255}$ and $\ell_2$ perturbations with radius 0.5. For PAT-VR, we use LPIPS computed from an AlexNet model (Krizhevsky et al., 2017) trained on CIFAR-10. We provide additional details about training procedure in Appendix C. We also provide results for additional source threat models such as StAdv and Recolor in Appendix E.10.

**Target threat models** We evaluate AT-VR on a variety of target threat models including, $\ell_p$ adversaries ($\ell_\infty$, $\ell_2$, and $\ell_1$ adversaries), spatially transformed adversary (StAdv) (Xiao et al., 2018), and Recolor adversary (Laidlaw & Feizi, 2019). For StAdv and Recolor threat models, we use the original bounds from (Xiao et al., 2018) and (Laidlaw & Feizi, 2019) respectively. For all other threat models, we specify the bound ($\epsilon$) within the figures in this section. We also provide evaluations on additional adversaries including Wasserstein, JPEG, elastic, and LPIPS-based attacks in Appendix E.7 for CIFAR-10 ResNet-18 models.

**Baselines** We remark that we are studying the setting where the *learner has already chosen a source threat model* and during testing the model is evaluated on a *strictly larger unknown* target. Because of this, for AT-VR experiments, we use standard PGD-AT (Madry et al., 2018) (AT-VR with $\lambda = 0$) as a baseline. For PAT-VR experiments, we use PAT (PAT-VR with $\lambda = 0$) as a baseline. We note that VR can be combined with other training techniques such as TRADES (Zhang et al., 2019) and provide results in Appendix E.9.

## 5.3 Performance of AT-VR across different imperceptible target threat models

We first investigate the impact of AT-VR on robust accuracy across different target threat models that are strictly larger than the source threat model used for training. To enforce this, we evaluate robust accuracy on a target threat model that is the union of the source with a different threat model. For

| Dataset | Architecture | Source | $\lambda$ | Clean acc | Source acc | Union with Source | | | | Union all |
|---|---|---|---|---|---|---|---|---|---|---|
| | | | | | | $\ell_\infty$ $\epsilon=\frac{12}{255}$ | $\ell_2$ $\epsilon=1$ | StAdv | Re-color | |
| CIFAR-10 | ResNet-18 | $\ell_2$ | 0 | **88.49** | 66.65 | 6.44 | 34.72 | 0.76 | 66.52 | 0.33 |
| CIFAR-10 | ResNet-18 | $\ell_2$ | 1 | 85.21 | **67.38** | **13.43** | **40.74** | **34.40** | **67.30** | **11.77** |
| CIFAR-10 | ResNet-18 | $\ell_\infty$ | 0 | **82.83** | 47.47 | 28.09 | **24.94** | 4.38 | 47.47 | 2.48 |
| CIFAR-10 | ResNet-18 | $\ell_\infty$ | 0.5 | 72.91 | **48.84** | **33.69** | 24.38 | **18.62** | **48.84** | **12.59** |
| CIFAR-10 | WRN-28-10 | $\ell_\infty$ | 0 | **85.93** | 49.86 | 28.73 | 20.89 | 2.28 | 49.86 | 1.10 |
| CIFAR-10 | WRN-28-10 | $\ell_\infty$ | 0.7 | 72.73 | **49.94** | **35.11** | **22.30** | **25.33** | **49.94** | **14.72** |
| CIFAR-10 | VGG-16 | $\ell_\infty$ | 0 | **79.67** | 44.36 | 26.14 | 30.82 | 7.31 | 44.36 | 4.35 |
| CIFAR-10 | VGG-16 | $\ell_\infty$ | 0.1 | 77.80 | **45.42** | **28.41** | **32.08** | **10.57** | **45.42** | **6.83** |
| ImageNette | ResNet-18 | $\ell_2$ | 0 | **88.94** | **84.99** | 0.00 | 79.08 | 1.27 | 72.15 | 0.00 |
| ImageNette | ResNet-18 | $\ell_2$ | 1 | 85.22 | 83.08 | **9.53** | **80.43** | **18.04** | **75.26** | **6.80** |
| ImageNette | ResNet-18 | $\ell_\infty$ | 0 | **80.56** | 49.63 | 32.38 | 49.63 | 34.27 | 49.63 | 25.68 |
| ImageNette | ResNet-18 | $\ell_\infty$ | 0.1 | 78.01 | **50.80** | **35.57** | **50.80** | **42.37** | **50.80** | **31.82** |
| CIFAR-100 | ResNet-18 | $\ell_2$ | 0 | **60.92** | 36.01 | 3.98 | 16.90 | 1.80 | 34.87 | 0.40 |
| CIFAR-100 | ResNet-18 | $\ell_2$ | 0.75 | 51.53 | **38.26** | **11.47** | **25.65** | **5.12** | **36.96** | **3.11** |
| CIFAR-100 | ResNet-18 | $\ell_\infty$ | 0 | **54.94** | 22.74 | 12.61 | 14.40 | 3.99 | 22.71 | 2.42 |
| CIFAR-100 | ResNet-18 | $\ell_\infty$ | 0.2 | 48.97 | **25.04** | **16.48** | **15.82** | **4.96** | **24.95** | **3.48** |

Table 1: Robust accuracy of various models trained at different strengths of VR applied on logits on various threat models. $\lambda = 0$ represents the baseline (standard AT). The "source acc" column reports the accuracy on the source attack ($\ell_\infty, \epsilon = \frac{8}{255}$ or $\ell_2, \epsilon = 0.5$). For each individual threat model, we evaluate accuracy on a union with the source threat model. The union all column reports the accuracy on the union across all listed threat models.

$\ell_\infty$ and $\ell_2$ attacks, we measure accuracy using AutoAttack (Croce & Hein, 2020a), which reports the lowest robust accuracy out of 4 different attacks: APGD-CE, APGD-T, FAB-T, and Square. For $\ell_\infty$ and $\ell_2$ threat models, we use radius $\epsilon = \frac{12}{255}$ and $\epsilon = 1$ for evaluating unforeseen robustness. We report clean accuracy, source accuracy (robust accuracy on the source threat model), and robust accuracy across various targets in Table 1. We present results with additional strengths of VR in Appendix E.5.

**AT-VR improves robust accuracy on broader target threat models.** We find that overall across datasets, architecture, and source threat model, using AT-VR improves robust accuracy on unforeseen targets but trades off clean accuracy. For instance, we find that on CIFAR-10, our ResNet-18 model using VR improves robustness on the union of all attacks from 2.48% to 12.59% for $\ell_\infty$ source and from 0.33% to 11.77% for $\ell_2$. The largest improvement we observe is a 33.64% increase in robust accuracy for the ResNet-18 CIFAR-10 with $\ell_2$ source model on the StAdv target.

**AT-VR maintains accuracy on the source compared to standard AT, but trades off clean accuracy.** We find that AT-VR is able to maintain similar source accuracy in comparson to standard PGD AT but consistently trades off clean accuracy. For example, for WRN-28-10 on CIFAR-10, we find that source accuracy increases slightly with VR (from 49.86% to 49.94%), but clean accuracy drops from 85.93% to 72.73%. In Appendix E.5, where we provide results on additional values of regularization strength ($\lambda$), *we find that increasing $\lambda$ generally trades off clean accuracy but improves union accuracy*. We hypothesize that this tradeoff occurs because VR enforces the decision boundary to be smooth, which may prevent the model from fitting certain inputs well.

### 5.4 State-of-the-art performance with PAT-VR

We now combine variation regularization with PAT. We present results in Table 2.

| Source $\epsilon$ | $\lambda$ | Clean acc | $\ell_\infty$ $\epsilon=\frac{8}{255}$ | $\ell_2$ $\epsilon=1$ | StAdv | Re-color | Union | PPGD | LPA |
|---|---|---|---|---|---|---|---|---|---|
| 0.5 | 0 | 86.6 | **38.8** | **44.3** | 5.8 | 60.8 | 2.1 | 16.2 | 2.2 |
| 0.5 | 0.05 | **86.9** | 34.9 | 40.6 | 9.4 | 64.6 | 3.7 | 21.9 | 2.2 |
| 0.5 | 0.1 | 85.1 | 31.4 | 37.1 | **44.9** | **80.5** | **24.9** | **48.7** | **29.7** |
| 1 [1] | 0 | 71.6 | 28.7 | 33.3 | **64.5** | 67.5 | 27.8 | 26.6 | 9.8 |
| 1 | 0.05 | 72.1 | **29.5** | 34.8 | 59.6 | 69.7 | 28.2 | 56.7 | 18.5 |
| 1 | 0.1 | **72.5** | 29.4 | **35.1** | 61.8 | **70.7** | **28.8** | **56.9** | **30.8** |

Table 2: Robust accuracy of ResNet-50 models trained using AlexNet-based PAT-VR with $\epsilon = 0.5$ and $\epsilon = 1$. $\lambda = 0$ corresponds to standard PAT. The union column reports the accuracy obtained on the union of $\ell_\infty$, $\ell_2$, StAdv, and Recolor adversaries. The PPGD and LPA columns report robust accuracy under AlexNet-based PPGD and LPA attacks with $\epsilon = 0.5$.

**PAT-VR achieves state-of-the-art robust accuracy on AlexNet-based LPIPS attacks (PPGD and LPA).** Laidlaw et al. (2021) observed that LPA attacks are the strongest perceptual attacks, and that standard AlexNet-based PAT with source $\epsilon = 1$ can only achieve 9.8% robust accuracy on LPA attacks with $\epsilon = 0.5$. In comparison, we find that applying variation regularization can significantly improve over performance on LPIPS attacks. In fact, using variation regularization strength $\lambda = 0.1$ while training with $\epsilon = 0.5$ can achieve 29.7% robust accuracy on LPA, while training with $\lambda = 0.1$ and $\epsilon = 1$ improves LPA accuracy to 30.8%.

**PAT-VR achieves state-of-the-art union accuracy across $\ell_\infty$, $\ell_2$, StAdv, and Recolor attacks.** We observe that as regularization strength $\lambda$ increases, union accuracy also increases. For source $\epsilon = 0.5$, we find that union accuracy increases from 2.1% without variation regularization to 24.9% with variation regularization at $\lambda = 0.1$. For source $\epsilon = 1$, we observe a 1% increase in union accuracy from $\lambda = 0$ to $\lambda = 0.1$. However, this comes at a trade-off with accuracy on specific threat models. For example, when training with $\epsilon = 0.5$, we find that variation regularization at $\lambda = 0.1$ trades off accuracy on $\ell_\infty$ and $\ell_2$ sources (7.4% and 7.2% drop in robust accuracy respectively), but improves robust accuracy on StAdv attacks from 5.8% to 44.9%. Meanwhile, for $\epsilon = 1$, we find that at $\lambda = 0.1$, variation regularization trades off accuracy on StAdv to improve accuracy across $\ell_\infty$, $\ell_2$, and Recolor threat models.

**Unlike AT-VR, PAT-VR maintains clean accuracy in comparison to PAT.** We find that PAT-VR generally does not trade off additional clean accuracy in comparison to PAT. In some cases (at source $\epsilon = 1$), increasing variation regularization strength can even improve clean accuracy.

## 5.5 Influence of AT-VR on threat model generalization gap across perturbation size

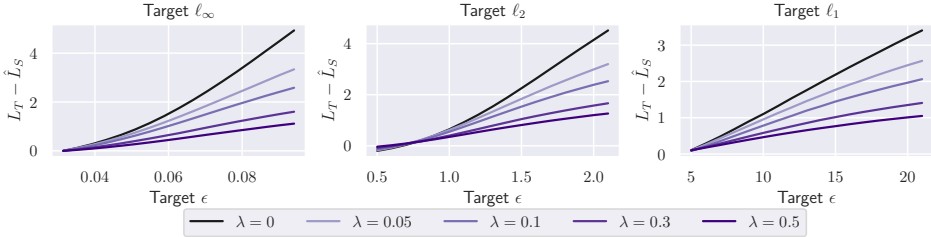

Figure 1: Threat model generalization gap of ResNet-18 models on CIFAR-10 trained using AT-VR at regularization strength $\lambda$ measured on target $\ell_p, p = \{\infty, 2, 1\}$ adversarial examples with radius $\epsilon$. The generalization gap is measured with respect to cross entropy loss. All models are trained with source $\ell_\infty$ perturbations of radius $\frac{8}{255}$. We find that increasing VR strength decreases the generalization gap across $\epsilon$.

In Section 5.3, we observed that AT-VR improves robust accuracy on a variety of unforeseen target threat models at the cost of clean accuracy. This suggests that AT-VR makes the change in adversarial loss on more difficult threat models increase more gradually. In this section, we experimentally verify this by plotting the gap between source and target losses (measured via cross entropy) across different perturbation strengths $\epsilon$ for $\ell_p$ threat models ($p \in \{\infty, 2, 1\}$) for ResNet-18 models trained on CIFAR-10. We present results for models using AT-VR with $\ell_\infty$ source attacks in Figure 1. For these experiments, we generate adversarial examples using APGD from AutoAttack (Croce & Hein, 2020b). We also provide corresponding plots for $\ell_2$ source attacks in Appendix E.4.

We find that AT-VR consistently reduces the gap between source and target losses on $\ell_p$ attacks across different target perturbation strengths $\epsilon$. *We observe that this gap decreases as regularization increases across target threat models*. This suggests that VR can reduce the generalization gap across threat models, making the loss measured on the source threat model better reflect the loss measured on the target threat model, which matches our results from Corollary 4.4.

## 5.6 Visualizing the expansion function

The effectiveness of AT-VR suggests that an expansion function exists between across the different imperceptible threat models tested. In this section, we visualize the expansion function between $\ell_\infty$

---

[1]Values taken from Laidlaw et al. (2021)

and $\ell_2$ source and target pairs for ResNet-18 models on CIFAR-10. We train a total of 15 ResNet-18 models using PGD-AT with and without VR on $\ell_2$ and $\ell_\infty$ source threat models. We evaluate variation on models saved every 10 epochs during training along with the model saved at epoch with best performance, leading to variation computation on a total of 315 models for each source threat model.

We consider 4 cases: (1) $\ell_\infty$ source with $\epsilon = \frac{8}{255}$ to $\ell_\infty$ target with $\epsilon = \frac{16}{255}$, (2) $\ell_\infty$ source with $\epsilon = \frac{8}{255}$ to a target consisting of the union of the source with an $\ell_2$ threat model with radius 0.5, (3) $\ell_2$ source with $\epsilon = 0.5$ to a target consisting of the union of the source with an $\ell_\infty$ threat model with $\epsilon = \frac{8}{255}$, and (4) $\ell_2$ source with $\epsilon = 0.5$ to $\ell_2$ target with $\epsilon = 1$. In cases (2) and (3), since the target is the union of $\ell_p$ balls, we approximate the variation of the union by taking the maximum variation across both $\ell_p$ balls. We plot the measured source vs target variation along with the minimum linear expansion function $s$ in Figure 2.

We find that in all cases the distribution of source vs target variation is sublinear, and we can upper bound this distribution with a linear expansion function with relatively small slope. Recall our finding in Theorem 4.7 that for linear models there exists a linear expansion function across $\ell_p$ norms. We hypothesize that this property also appears for ResNet-18 models because neural networks are piecewise linear.

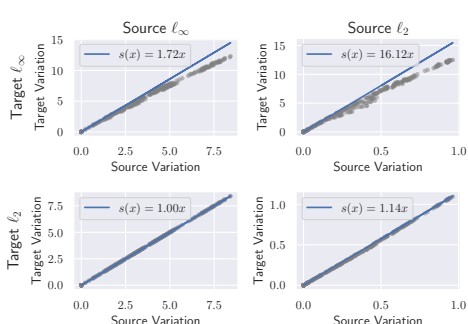

Figure 2: Plots of minimum linear expansion function $s$ shown in blue computed on 315 adversarially trained ResNet-18 models. Each grey point represents variation measured on the source and target pair. Variation is computed on the logits. The two columns represent the source adversary ($\ell_\infty$ and $\ell_2$ respectively). The two rows represent the target adversary ($\ell_\infty$ and $\ell_2$ respectively).

## 6 Discussion, Limitations, and Conclusion

We highlight a limitation in adversarial ML research: the lack of understanding of how robustness degrades when a mismatch in source and target threat models occurs. Our work takes steps toward addressing this problem by formulating the problem of learning with an unforeseen adversary and providing a framework for reasoning about generalization under this setting. With this framework, we derive a bound for threat model generalization gap in terms of variation and use this bound to design an algorithm, adversarial training with variation regularization (AT-VR). We highlight several limitations of our theoretical results: (1) the bounds provided can be quite loose and may not be good predictors of unforeseen loss, (2) while we show that an expansion function between $\ell_p$ balls exists for linear models, it is unclear if that is the case for neural networks. Additionally, we highlight several limitations of AT-VR: (1) its success depends on the existence of an expansion function, (2) VR trades off additional clean accuracy and increases computational complexity of training. Further research on improving source threat models and the accuracy and efficiency of adversarial training algorithms can improve the performance of AT-VR. Finally, we note that in some applications, such as defending against website fingerprinting (Rahman et al., 2020) and bypassing facial recognition based surveillance (Shan et al., 2020), adversarial examples are used for good, so improving robustness against adversarial examples may consequently hurt these applications.

## Acknowledgments and Disclosure of Funding

We would like to thank Tianle Cai, Peter Ramadge, and Vincent Poor for their feedback on this work. This work was supported in part by the National Science Foundation under grants CNS-1553437 and CNS-1704105, the ARL's Army Artificial Intelligence Innovation Institute (A2I2), the Office of Naval Research Young Investigator Award, the Army Research Office Young Investigator Prize, Schmidt DataX award, and Princeton E-ffiliates Award. This material is also based upon work supported by the National Science Foundation Graduate Research Fellowship under Grant No. DGE-2039656. Any opinions, findings, and conclusions or recommendations expressed in this material are those of the author(s) and do not necessarily reflect the views of the National Science Foundation.

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
