# Appendix

## Table of Contents

## A  Discussion of Related Works

**Comparison to Ye et al. (2021)**    Ye et al. (2021) derive a generalization bound for the domain generalization problem in terms of variation across features. They define variation as $\mathcal{V}_\rho(\phi, \mathcal{D}) = \max_{y \in \mathcal{Y}} \sup_{e,e' \in \mathcal{D}} \rho(\mathbb{P}(\phi(X_e)|y), \mathbb{P}(\phi(X_{e'})|y))$ where $\mathcal{D}$ is a set of training distributions, $\phi : \mathcal{X} \rightarrow \mathbb{R}$ is a function that maps the input to a 1-D feature, $\rho$ is a symmetric distance metric for distributions, and $X_e$ denotes inputs from domain $e$. In comparison, we address the problem of learning with an unforeseen adversary and define unforeseen adversarial generalizability. Our generalization bound using variation is an instantiation of our generalizability framework. Additionally, we define a different measure of variation for threat models $\mathcal{V}(h, N) = \mathbb{E}_{(x,y) \sim \mathcal{D}} \max_{x_1, x_2 \in N(x)} ||h(x_1), -h(x_2)||_2$, which allows us to use it as regularization during training.

**Comparison to Laidlaw et al. (2021)** Laidlaw et al. (2021) proposes training with perturbations of bounded LPIPS (Zhang et al., 2018) since LPIPS metric is a better approximation of human perceptual distance than $\ell_p$ metrics. In their proposed algorithm, perceptual adversarial training (PAT), they combine standard adversarial training with adversarial examples generated via their LPIPS bounded attack method. In terms of terminology introduced in our paper, the Laidlaw et al. (2021) improve the choice of source threat model while using an existing learning algorithm (adversarial training). Meanwhile, our work takes the perspective of having a fixed source threat model and improving the learning algorithm. This allows us to combine our approach with various source threat models including the attacks used by Laidlaw et al. (2021) in PAT (see Appendix E.10).

**Comparison to Stutz et al. (2020) and Chen et al. (2022)** Stutz et al. (2020) and Chen et al. (2022) address the problem of unforeseen attacks by adding a reject option in order to reject adversarial examples generated with a larger perturbation than used during training. These techniques introduce a modified adversarial training objective that maximizes accuracy on perturbations within the source threat model and maximize rejection rate of large perturbations. In comparison, we look at the problem of improving robustness on larger threat models instead of rejecting adversarial examples from larger threat models. Our algorithm AT-VR actively tries to find a robust model that minimizes our generalization bound without abstaining on any inputs.

**Comparison to Croce & Hein (2020a)** Croce & Hein (2020a) prove that certified robustness against $\ell_1$ and $\ell_\infty$ bounded perturbations implies certified robustness against attacks generated with any $\ell_p$ ball. The size of this $\ell_p$ certified radius is the radius of the largest $\ell_p$ ball that can be contained in the convex hull of the $\ell_1$ and $\ell_\infty$ balls for which the model is certifiably robust. In our work, we are primarily interested in empirical robustness on target threat models that are supersets of the source threat model used. We demonstrate that by using variation regularization, we can improve robust performance on unforeseen threat models (including larger $\ell_p$ perturbations, StAdv, and Recolor) even when our learning algorithm optimizes for robust models on a single $\ell_p$ ball.

**Comparison to other forms of regularization for adversarial robustness** Prior works in adversarial training propose regularization techniques enforcing feature consistency to improve the performance of adversarial training. For example, TRADES adversarial training(Zhang et al., 2019) uses a regularization term in the objective to reduce trade-off between clean accuracy and robust accuracy compared with PGD adversarial training. This regularization term takes the form: $\lambda \max_{\hat{x} \in S(x)} \ell(f(x)f(\hat{x}))$. Our variation regularization differs from TRADES since we regularize $\ell_2$ distance between extracted features.

Another regularization is adversarial logit pairing (ALP) (Kannan et al., 2018). This regularization is also $\ell_2$ based; in ALP, an adversarial example is first generated via $x' = \max_{\hat{x} \in S(x)} \ell(f(\hat{x}), y)$ and the $\ell_2$ distance between the logits of this adversarial example and the original image $\lambda ||f(x') - f(x)||_2$ is added to the training objective. ALP can be thought of as a technique to make the logits of adversarial examples close to the logits of clean images. Variation regularization ($\lambda \max_{x',x'' \in S(x)} ||f(x') - f(x'')||_2$) differs from ALP since it encourages the logits of any pair of images (not only adversarial examples) that lie within the source threat model to have similar features and does not use information about the label of the image.

Jin & Rinard (2020) propose a regularization technique motivated by concepts from manifold regularization. Their regularization is computed with randomly sampled maximal perturbations $p \in \{\pm \epsilon\}^d$ and applied to standard training. Their regularization includes 2 terms, one which regularizes the hamming distance between the ReLU masks of the network for $x - p$ and $x + p$ across inputs $x$, and the second which regularizes the squared $\ell_2$ distance between the network output on $x - p$ and $x + p$ ($||f(x + p) - f(x - p)||_2^2$). They demonstrate that using both regularization terms with $\epsilon \in [2, 8]$ leads to robustness on $\ell_\infty$, $\ell_2$, and Wasserstein attacks. In comparison, our variation regularization is motivated from the perspective of generalization across threat models. We use use smaller values of $\epsilon$ in conjunction with adversarial training and regularize worst case $\ell_2$ distance between logits of any pair of examples within the source threat model.

# B  Proofs of Theorems and Analysis of Bounds

## B.1  Proof of Theorem 4.2

*Proof.* By definition of expected adversarial risk, we have that for any $f \in \mathcal{F}$

$$L_T(f) - L_S(f) = \mathbb{E}_{(x,y) \sim \mathcal{D}}(\max_{x_1 \in T(x)} \ell(f(x_1), y) - \max_{x_2 \in S(x)} \ell(f(x_2), y))$$

$$\leq \mathbb{E}_{(x,y) \sim \mathcal{D}} \max_{x_1 \in T(x), x_2 \in S(x)} (\ell(f(x_1), y) - \ell(f(x_2), y))$$

By $\rho$-Lipschitzness of $\ell$

$$\leq \mathbb{E}_{(x,y) \sim \mathcal{D}} \max_{x_1 \in T(x), x_2 \in S(x)} \rho ||f(x_1) - f(x_2)||_2$$

$$= \rho \mathbb{E}_{(x,y) \sim \mathcal{D}} \max_{x_1 \in T(x), x_2 \in S(x)} ||g(h(x_1)) - g(h(x_2))||_2$$

By $\sigma_\mathcal{G}$ Lipschitzness:

$$\leq \rho \sigma_\mathcal{G} \mathbb{E}_{(x,y) \sim \mathcal{D}} \max_{x_1 \in T(x), x_2 \in S(x)} ||h(x_1) - h(x_2)||_2$$

Since $S \subseteq T$:

$$\leq \rho \sigma_\mathcal{G} \mathbb{E}_{(x,y) \sim \mathcal{D}} \max_{x_1, x_2 \in T(x)} ||h(x_1) - h(x_2)||_2$$

$$= \rho \sigma_\mathcal{G} \mathcal{V}(h, T)$$

Note that our learning algorithm $\mathcal{A}$ over $\mathcal{F}$ outputs a classifier with $\mathcal{V}(h, T) \leq \epsilon(T, m)$ with probability $1 - \delta$. Combining this with the above bound gives:

$$\mathbb{P}[L_T(f) - L_S(f) \leq \rho \sigma_\mathcal{G} \epsilon(T, m)] \geq 1 - \delta$$

$\square$

## B.2  Impact of choice of source and target

In Theorem 4.2, we demonstrated a bound on threat model generalization gap that does not depend on source threat model, so this bound does not allow us to understand what types of targets are easier to generalize given a source threat model. In this section, we will introduce a tighter bound in terms of Hausdorff distance, which takes both source and target threat model into account.

**Definition B.1** (Directed Hausdorff Distance). Let $A, B \subset X$ and let $d : X \times X \to \mathbb{R}$ be a distance metric. The Hausdorff distance from $A$ to $B$ based on $d$ is given by

$$H_d(A, B) = \max_{x_1 \in A} \min_{x_2 \in B} d(x_1, x_2)$$

Intuitively what this measures is, if we were to take every point in $A$ and project it to the nearest point in set $B$, what is the maximum distance projection? We can then derive a generalization bound in terms of Hausdorff distance based on feature space distance.

**Theorem B.2** (Threat Model Generalization Bound with Hausdorff Distance). *Let $S$ denote the source threat model and $\mathcal{D}$ denote the data distribution. Let $\mathcal{F} = \mathcal{G} \circ \mathcal{H}$. Let $\mathcal{G}$ be a class of Lipschitz classifiers, where the Lipschitz constant is upper bounded by $\sigma_\mathcal{G}$. Let $\ell$ be a $\rho$-Lipschitz loss function with respect to the 2-norm. Then, for any target threat model $T$ with $S \subseteq T$. Let $d_h := ||h(x_1) - h(x_2)||_2$ be the distance between feature extracted by the model. Then,*

$$L_T(f) - L_S(f) \leq \rho \sigma_\mathcal{G} \mathbb{E}_{(x,y) \sim \mathcal{D}}[H_{d_h}(T(x), S(x))]$$

*Proof.* By definition of expected adversarial risk,

$$L_T(f) - L_S(f) = \mathbb{E}_{(x,y) \sim \mathcal{D}}(\max_{x_1 \in T(x)} \ell(f(x_1), y) - \max_{x_2 \in S(x)} \ell(f(x_2), y))$$

Note that since we are subtracting the max across $S(x)$, this expression is upper bounded by any choice of $\hat{x} \in S(x)$. Thus, we can choose $\hat{x}$ so that $\ell(f(\hat{x}), y)$ is close to $\max_{x_1 \in T(x)} \ell(f(x_1), y$. This gives us:

$$\leq \mathbb{E}_{(x,y) \sim \mathcal{D}}(\max_{x_1 \in T(x)} \min_{\hat{x} \in S(x)} \ell(f(x_1), y) - \ell(f(\hat{x}), y))$$

By $\rho$-Lipschitzness of $\ell$

$$\leq \mathbb{E}_{(x,y)\sim\mathcal{D}} \max_{x_1\in T(x)} \min_{x_2\in S(x)} \rho||f(x_1) - f(x_2)||_2$$

$$= \rho\mathbb{E}_{(x,y)\sim\mathcal{D}} \max_{x_1\in T(x)} \min_{x_2\in S(x)} ||g(h(x_1)) - g(h(x_2))||_2$$

By $\sigma_\mathcal{G}$ Lipschitzness:

$$\leq \rho\sigma_\mathcal{G}\mathbb{E}_{(x,y)\sim\mathcal{D}} \max_{x_1\in T(x)} \min_{x_2\in S(x)} ||h(x_1) - h(x_2)||_2$$

Since $S \subseteq T$:

$$\leq \rho\sigma_\mathcal{G}\mathbb{E}_{(x,y)\sim\mathcal{D}} \max_{x_1\in T(x)} \min_{x_2\in S(x)} ||h(x_1) - h(x_2)||_2$$

$$= \rho\sigma_\mathcal{G}\mathbb{E}_{(x,y)\sim\mathcal{D}}[H_{d_h}(T(x), S(x))]$$

$\square$

We note that this bound is tighter than the variation based bound and goes to 0 when $S = T$. Since this bound also depends on both $S$ and $T$, we can also see that the "difficulty" of a target $T$ with respect to a chosen source threat model $S$ can be measured through the directed Hausdorff distance from $T(x)$ to $S(x)$.

### B.3   Proof of Theorem 4.7

**Lemma B.3** (Variation Upper Bound for $\ell_p$ threat model, $p \in \mathbb{N} \cup +\infty$)**.** *Let inputs $x \in \mathbb{R}^n$ and corresponding label $y \in [1...K]$. Let the adversarial constraint be given by $T(x) = \{\hat{x}|\ ||\hat{x} - x||_p \leq \epsilon_{\max}\}$ Let $h$ be a linear feature extractor: $h \in \{Wx + b|W \in \mathbb{R}^{d\times n}, b \in \mathbb{R}^d\}$. Then, variation is upper bounded by*

$$\mathcal{V}_p(h, T) \leq \begin{cases} 2\epsilon_{\max}n^{\frac{1}{2}-\frac{1}{p}}\sigma_{\max}(W) & p \geq 2 \\ 2\epsilon_{\max}\sigma_{\max}(W) & p = 1, p = 2 \end{cases}$$

*Proof.*

$$\mathcal{V}(h, T) = \mathbb{E}_{(x,y)\sim D} \max_{x_1,x_2\in T(x)} ||h(x_1) - h(x_2)||_2$$

$$= \mathbb{E}_{(x,y)\sim D} \max_{x_1,x_2\in T(x)} ||W(x_1 - x_2)||_2$$

$$\leq \mathbb{E}_{(x,y)\sim D} \max_{x_1,x_2\in T(x)} \sigma_{\max}(W)||x_1 - x_2||_2 \tag{1}$$

Consider the case where $p > 2$. Then, by Hölder's inequality:

$$\leq \mathbb{E}_{(x,y)\sim D} \max_{x_1,x_2\in T(x)} n^{\frac{1}{2}-\frac{1}{p}}\sigma_{\max}(W)||x_1 - x_2||_p$$

$$\leq 2\epsilon_{\max}n^{\frac{1}{2}-\frac{1}{p}}\sigma_{\max}(W)$$

When $p = 1$ or $p = 2$, then from 1, we have:

$$\leq \mathbb{E}_{(x,y)\sim D} \max_{x_1,x_2\in T(x)} \sigma_{\max}(W)||x_1 - x_2||_p$$

$$\leq 2\epsilon_{\max}\sigma_{\max}(W)$$

$\square$

**Lemma B.4** (Variation lower bound for $\ell_p$ threat model, $p \in \mathbb{N} \cup +\infty$)**.** *Let inputs $x \in \mathbb{R}^n$ and corresponding label $y \in [1...K]$. Let the adversarial constraint be given by $T(x) = \{\hat{x}|\ ||\hat{x} - x||_p \leq \epsilon_{\max}\}$. Let $h$ be a linear feature extractor: $h \in \{Wx + b|W \in \mathbb{R}^{d\times n}, b \in \mathbb{R}^d\}$. Then, variation is lower bounded by*

$$\mathcal{V}_p(h, T) \geq \begin{cases} 2\epsilon_{\max}\sigma_{\min}(W) & p \geq 2 \\ \frac{2\epsilon_{\max}}{\sqrt{n}}\sigma_{\min}(W) & p = 1 \end{cases}$$

*where $\sigma_{\min}(W)$ denotes the smallest singular value of $W$.*

*Proof.*

$$\mathcal{V}(h,T) = \mathbb{E}_{(x,y)\sim D} \max_{x_1,x_2\in T(x)} ||W(x_1-x_2)||_2$$

$$\geq \sigma_{\min}(W)\mathbb{E}_{(x,y)\sim D} \max_{x_1,x_2\in T(x)} ||x_1-x_2||_2 \tag{2}$$

Then, for $p \geq 2$:

$$\geq \sigma_{\min}(W)\mathbb{E}_{(x,y)\sim D} \max_{x_1,x_2\in T(x)} ||x_1-x_2||_p$$

$$= 2\epsilon_{\max}\sigma_{\min}(W)$$

For $p = 1$ from 2, we have:

$$\geq \frac{1}{\sqrt{n}}\sigma_{\min}(W)\mathbb{E}_{(x,y)\sim D} \max_{x_1,x_2\in T(x)} ||x_1-x_2||_1$$

$$= \frac{2\epsilon_{\max}}{\sqrt{n}}\sigma_{\min}(W)$$

$\square$

**Lemma B.5** ($\ell_p$ threat models with larger radius, $p \in \mathbb{N} \cup +\infty$)**.** *Let inputs $x \in \mathbb{R}^n$ and corresponding label $y \in [1...K]$. Let $S(x) = \{\hat{x}| ||\hat{x}-x||_p \leq \epsilon_1\}$ and $T(x) = \{\hat{x}| ||\hat{x}-x||_p \leq \epsilon_2\}$ where $\epsilon_2 \geq \epsilon_1$. Consider a linear feature extractor with bounded condition number: $h \in \{Wx+b | W \in \mathbb{R}^{d\times n}, b \in \mathbb{R}^d, \frac{\sigma_{\max}(W)}{\sigma_{\min}(W)} \leq B < \infty\}$. Then a valid expansion function is given by:*

$$s_p(z) = \begin{cases} \sqrt{n}B\frac{\epsilon_2}{\epsilon_1}z & p = 1 \\ B\frac{\epsilon_2}{\epsilon_1}z & p = 2 \\ n^{\frac{1}{2}-\frac{1}{p}}B\frac{\epsilon_2}{\epsilon_1}z & p > 2 \end{cases}$$

*Proof.* By Lemma B.3 and Lemma B.4, $\mathcal{V}(h,T) \leq s(\mathcal{V}(h,S))$. Additionally, it is clear that $s_p$ satisfies properties of expansion function. $\square$

**Lemma B.6** (Variation upper bound for the union of $\ell_p$ and $\ell_q$ threat models ($p,q \in \mathbb{N} \cup +\infty$))**.** *Let inputs $x \in \mathbb{R}^n$ and corresponding label $y \in [1...K]$. Consider $T_1(x) = \{\hat{x}| ||\hat{x}-x||_p \leq \epsilon_1\}$ and $T_2(x) = \{\hat{x}| ||\hat{x}-x||_q \leq \epsilon_2\}$. Define adversarial constraint $T = T_1 \cup T_2$. Let $h$ be a linear feature extractor: $h \in \{Wx+b | W \in \mathbb{R}^{d\times n}, b \in \mathbb{R}^d\}$. Let $v(p,h,T)$ be defined as*

$$v(p,\epsilon,W) = \begin{cases} 2\epsilon n^{\frac{1}{2}-\frac{1}{p}}\sigma_{\max}(W) & p \geq 2 \\ 2\epsilon\sigma_{\max}(W) & p = 1, p = 2 \end{cases}$$

*where $\sigma_{\max}(W)$ denotes the largest singular value of $W$.*

*Then variation is upper bounded by:*

$$\mathcal{V}_{(p,\epsilon_1),(q,\epsilon_2)}(h,T) \leq \max(v(p,\epsilon_1,W), v(q,\epsilon_2,W))$$

*Proof.*

$$\mathcal{V}(h,T) = \mathbb{E}_{(x,y)\sim D} \max_{x_1,x_2\in T(x)} ||W(x_1-x_2)||_2$$

$$\leq \mathbb{E}_{(x,y)\sim D} \max_{x_1,x_2\in T(x)} \sigma_{\max}(W)||x_1-x_2||_2$$

Since $T = T_1 \cup T_2$, $\max_{x_1,x_2\in T(x)} ||x_1-x_2||_2$ can be upper bounded by the diameter of the hypersphere that contains both $T_1$ and $T_2$. We can compute this diameter by taking the max out of the diameter of the hypersphere containing $T_1$ and the diameter of the hypersphere containing $T_2$. This was computed in proof of Lemma B.3 to bound the case of a single $\ell_p$ norm. Thus,

$$\mathcal{V}_{(p,\epsilon_1),(q,\epsilon_2)}(h,T) \leq \max(v(p,\epsilon_1,W), v(q,\epsilon_2,W))$$

where the expression for $v$ follows from the result of Lemma B.3. $\square$

**Lemma B.7** ($\ell_p$ to union of $\ell_p$ and $\ell_q$ threat model ($p, q \in \mathbb{N} \cup +\infty$) ). *Let inputs $x \in \mathbb{R}^n$ and corresponding label $y \in [1...K]$. Consider $S(x) = \{\hat{x} \mid ||\hat{x} - x||_p \le \epsilon_1\}$ and $U(x) = \{\hat{x} \mid ||\hat{x} - x||_q \le \epsilon_2\}$. Define target threat model $T = S \cup U$. Consider a linear feature extractor with bounded condition number: $h \in \{Wx + b \mid W \in \mathbb{R}^{d \times n}, b \in \mathbb{R}^d, \frac{\sigma_{\max}(W)}{\sigma_{\min}(W)} \le B < \infty\}$. Then a valid expansion function is given by:*

$$
s_{p,q}(z) = \begin{cases}
\sqrt{n} B \frac{\max(\epsilon_2, \epsilon_1)}{\epsilon_1} & p = 1, q = 2 \\
\sqrt{n} B \frac{\max(n^{\frac{1}{2} - \frac{1}{q}} \epsilon_2, \epsilon_1)}{\epsilon_1} & p = 1, q > 2 \\
B \frac{\max(\epsilon_2, \epsilon_1)}{\epsilon_1} & p = 2, q = 1 \\
B \frac{\max(n^{\frac{1}{2} - \frac{1}{q}} \epsilon_2, \epsilon_1)}{\epsilon_1} & p = 2, q > 2 \\
B \frac{\max(\epsilon_2, n^{\frac{1}{2} - \frac{1}{p}} \epsilon_1)}{\epsilon_1} & p > 2, q \le 2 \\
B \frac{\max(n^{\frac{1}{2} - \frac{1}{q}} \epsilon_2, n^{\frac{1}{2} - \frac{1}{p}} \epsilon_1)}{\epsilon_1} & p > 2, q > 2
\end{cases}
$$

*Proof.* By Lemma B.6 and Lemma B.4, $\mathcal{V}(h, T) \le s(\mathcal{V}(h, S))$. Additionally, it is clear that $s_p$ satisfies properties of expansion function. □

*Proof of Theorem 4.7.* Directly follows from Lemma B.5 and Lemma B.7. □

## B.4 How well can empirical expansion function predict loss on the target threat model for neural networks?

Using the empirical expansion function slopes from Figures 2 and 8, we compute the expected cross entropy loss (with softmax) on the target threat model via Corollary 4.4. We provide the predicted and true target losses in Table 3 for $\ell_\infty, \epsilon = \frac{8}{255}$ source threat model and 4 for $\ell_2, \epsilon = 0.5$.

| Target | Source Variation | Source Loss | Predicted Target loss | True Target Loss | Gap |
|---|---|---|---|---|---|
| $\ell_\infty, \epsilon = \frac{16}{255}$ | 4.90 | 0.93 | 12.85 | 2.44 | 10.41 |
| $\ell_2, \epsilon = 0.5$ | 4.90 | 0.93 | 7.86 | 0.93 | 6.93 |
| StAdv $\epsilon = 0.05$ | 4.90 | 0.93 | 9.87 | 5.13 | 4.74 |
| $\ell_\infty, \epsilon = \frac{16}{255}$ | 0.98 | 1.26 | 3.64 | 1.76 | 1.88 |
| $\ell_2, \epsilon = 0.5$ | 0.98 | 1.26 | 2.64 | 1.27 | 1.37 |
| StAdv $\epsilon = 0.05$ | 0.98 | 1.26 | 3.05 | 2.11 | 0.94 |

Table 3: Predicted and measured losses on multiple target threat models for ResNet-18 model trained on CIFAR-10 with $\ell_\infty, \epsilon = 8/255$ source threat model.

| Target | Source Variation | Source Loss | Predicted Target loss | True Target Loss | Gap |
|---|---|---|---|---|---|
| $\ell_\infty, \epsilon = \frac{8}{255}$ | 0.78 | 0.64 | 18.52 | 2.65 | 15.87 |
| $\ell_2, \epsilon = 1$ | 0.78 | 0.64 | 1.91 | 1.93 | 0.02 |
| StAdv $\epsilon = 0.05$ | 0.78 | 0.64 | 16.51 | 12.26 | 4.25 |
| $\ell_\infty, \epsilon = \frac{8}{255}$ | 0.20 | 0.85 | 5.38 | 1.77 | 3.61 |
| $\ell_2, \epsilon = 1$ | 0.20 | 0.85 | 1.16 | 1.51 | 0.35 |
| StAdv $\epsilon = 0.05$ | 0.20 | 0.85 | 4.88 | 2.10 | 2.78 |

Table 4: Predicted and measured losses on multiple target threat models for ResNet-18 model trained on CIFAR-10 with $\ell_2, \epsilon = 0.5$ source threat model.

In general, we find that for models with smaller variation (those trained with variation regularization), the loss estimate using the slope from the expansion function generally improves. In the case where the target threat model is Linf, we believe that the large gap between predicted and true loss for the unregularized model stems from the fact that we model the expansion function with a linear model. From Figure 2, we can see that at larger values of source variation the linear model for expansion function becomes an increasingly loose upper-bound. Improving the model for expansion function (ie. using a log function) may reduce this gap.

## C   Additional Experimental Setup Details

**Variation Computation Algorithm for AT-VR**   We provide the algorithm we use to compute variation regularization for $\ell_p$ source adversaries in Algorithm 1.

---

**Algorithm 1:** Variation Regularization Computation for $\ell_p$ ball

---

**Input**      : $x \in \mathcal{X}$, $\ell_p$ radius $\epsilon$, $n$ number of steps, $\alpha$ step size, feature extractor $h$
**Notations :**$\mathcal{U}$ denotes the uniform distribution of dimension of the input, $\prod_{\ell_p,\epsilon}$ denotes
             projection onto $\ell_p$ ball of radius $\epsilon$
**Output**    : Variation $v \in \mathbb{R}$

1  $x_1 \leftarrow \prod_{\ell_p,\epsilon}(x + \mathcal{U}(-\epsilon,\epsilon))$ ;                    // randomly initialize $x_1$
2  $x_2 \leftarrow \prod_{\ell_p,\epsilon}(x + \mathcal{U}(-\epsilon,\epsilon))$ ;                    // randomly initialize $x_1$
3  **for** $i = 1...n$ **do**
4   $\quad$ $v \leftarrow ||h(x_1) - h(x_2)||_2$ ;                                      // compute objective
5   $\quad$ $x_1 \leftarrow \prod_{\ell_p,\epsilon}(x_1 + \alpha\nabla_{x_1}v)$ ;            // single step of PGD to optimize $x_1$
6   $\quad$ $x_2 \leftarrow \prod_{\ell_p,\epsilon}(x_2 + \alpha\nabla_{x_2}v)$ ;            // single step of PGD to optimize $x_2$
7  **end**
8  $v \leftarrow ||h(x_1) - h(x_2)||_2$ ;                                      // compute final variation
9  **return** $v$

---

**Variation Computation for PAT-VR**   Laidlaw et al. (2021) propose an algorithm called Fast Lagrangian Perceptual Attack (Fast-LPA) to generate adversarial examples for training using PAT. We make a slight modification of the Fast-LPA algorithm, replacing the loss in the original optimization objective with the variation objective, to obtain Fast Lagrange Perceptual Variation (Fast-LPV). The explicit algorithm for Fast-LPV is provided in Algorithm 2. We use the variation obtained from Fast-LPV for training with PAT-VR.

---

**Algorithm 2:** Fast Lagrangian Perceptual Variation (Fast-LPV)

---

**Input**      : feature extractor $h(\cdot)$, LPIPS distance $d(\cdot,\cdot)$, input $x$, label $y$, bound $\epsilon$, number of
             steps $n$
**Output**    : variation $v$

1  $x_1 \leftarrow x + 0.01 * \mathcal{N}(0.1)$ ;                              // randomly initialize $x_1$
2  $x_2 \leftarrow x + 0.01 * \mathcal{N}(0.1)$ ;                              // randomly initialize $x_2$
3  **for** $i = 1...n$ **do**
4   $\quad$ $\tau \leftarrow 10^{i/n}$ ;                                       // $\tau$ increases exponentially
5   $\quad$ $obj \leftarrow ||h(x_1) - h(x_2)||_2 - \tau(\max(0, d(x_1,x) - \epsilon) + \max(0, d(x_2,x) - \epsilon))$ ;    // obj
     represents optimization objective
6   $\quad$ $\Delta_1 \leftarrow \frac{\nabla_{x_1}[obj]}{||\nabla_{x_1}[obj]||_2}$ ;                       // compute normalized gradient wrt $x_1$
7   $\quad$ $\Delta_2 \leftarrow \frac{\nabla_{x_2}[obj]}{||\nabla_{x_2}[obj]||_2}$;                       // compute normalized gradient wrt $x_2$
8   $\quad$ $\eta \leftarrow \epsilon * (0.1)^{i/n}$ ;                              // step size $\eta$ decays exponentially
9   $\quad$ $m_1 \leftarrow d(x_1, x_1 + 0.1\Delta_1)/0.1$ ;                    // derivative of $d$ in direction of $\Delta_1$
10  $\quad$ $m_2 \leftarrow d(x_2, x_2 + 0.1\Delta_2)/0.1$ ;                    // derivative of $d$ in direction of $\Delta_2$
11  $\quad$ $x_1 \leftarrow x_1 + (\eta/m)\Delta_1$ ;                    // take a step of size $\eta$ in LPIPS distance
12  $\quad$ $x_2 \leftarrow x_2 + (\eta/m)\Delta_2$ ;                    // take a step of size $\eta$ in LPIPS distance
13 **end**
14 $v \leftarrow ||h(x_1) - h(x_2)||_2$;
15 **return** $v$

---

**Additional Experimental Setup Details for AT-VR**   We run all training on a NVIDIA A100 GPU. For all datasets and architectures, we perform PGD adversarial training (Madry et al., 2018) and add variation regularization to the objective. For all datasets, train with normalized inputs. On ImageNette, we normalize using ImageNet statistics and resize all images to $224 \times 224$. We train models on seed

0 for 200 epochs with SGD with initial learning rate of 0.1 and decrease the learning rate by a factor of 10 at the 100th and 150th epoch. We use 10-step PGD and train with $\ell_\infty$ perturbations of radius $\frac{8}{255}$ and $\ell_2$ perturbations with radius 0.5. For $\ell_\infty$ perturbations we use step size of $\alpha = \frac{2}{255}$ while for $\ell_2$ perturbations we use step size of 0.075. We use the same settings for computing the variation regularization term. For all models, we evaluate performance at the epoch which achieves the highest robust accuracy on the test set.

**Additional Experimental Setup Details for PAT-VR**   We build off the official code repository for PAT and train ResNet-50 models on CIFAR-10 with AlexNet-based LPIPS threat model with bound $\epsilon = 0.5$ and $\epsilon = 1$. For computing AlexNet-based LPIPS we use the CIFAR-10 AlexNet pretrained model provided in the PAT official code repository. We train all models for 100 epochs and evaluate the model saved at the final epoch of training. To match evaluation technique as Laidlaw et al. (2021), we evaluate with $\ell_\infty$ attacks, $\ell_2$ attacks, StAdv, and Recolor with perturbation bounds $\frac{8}{255}$, 1, 0.05, 0.06 respectively. Additionally, we present accuracy measured using AlexNet-based PPGD and LPA attacks (Laidlaw et al., 2021) with $\epsilon = 0.5$.

# D   Experiments for linear models on Gaussian data

In Section 4.3, we demonstrated that for a linear feature extractor, the expansion function exists, so decreasing variation across an $\ell_p$ source adversarial constraint should improve generalization to other $\ell_p$ constraints. We now verify this experimentally by training a linear model for binary classification on isotropic Gaussian data.

## D.1   Experimental Setup

**Data generation**   We sample data from 2 isotropic Gaussians with covariance $\sigma^2 I_n$ where $I_n$ denotes the $n \times n$ identity matrix. For class 0, we sample from a Gaussian with mean $\theta_0 = (0.25, 0, 0, ..., 0)^T$, and for class 1, we sample from a Gaussian with mean $\theta_1 = (0.75, 0, 0, ..., 0)^T$ and clip all samples to range $[0, 1]$ to simulate image data. We sample 1000 points per class. We vary $\sigma \in \{0.125, 0.25\}$ and $n \in \{25, 100\}$. Since our generalization bound considers only threat model generalization gap and not sample generalization gap, we evaluate the models using the same data samples as used during training for the bulk of our experiments, but we provide results on a separate test set for one setting in Appendix D.5.

**Model architecture**   We train a model consisting of a linear feature extractor and linear top level classifier: $f = g \circ h$ where $h(x) = Wx + b_1$ where $W \in \mathbb{R}^{n \times d}$, $b_1 \in \mathbb{R}^d$ and $g(x) = Ax + b_2$ where $A \in \mathbb{R}^{d \times 2}$, $b_2 \in \mathbb{R}^2$. For our experiments, we use $d \in \{5, 25\}$.

**Source Threat Models**   Across experiments we use 2 different source threat models: $\ell_\infty$ perturbations with radius 0.01 and $\ell_2$ perturbations with radius 0.01.

**Training Details**   We perform AT-VR with adversarial examples generated using APGD (Croce & Hein, 2020b) for 200 epochs. We train models using SGD with learning rate of 0.1 and momentum of 0.9. We use cross entropy loss during both training and evaluation. For variation regularization, we use 10 steps for optimization and use step size $\epsilon/9$ where $\epsilon$ is the radius of the $\ell_p$ ball used during training/evaluation.

## D.2   Visualizing the expansion function for Gaussian data

In Section 4.3, we demonstrated that with a linear feature extractor, for any dataset there exists a linear expansion function. We now visualize the true expansion function for Gaussian data with a linear function class across 4 different combinations of input dimension $n$, Gaussian standard deviation $\sigma$, and feature dimension $d$. We set our source threat model to be $\ell_p, p \in \{\infty, 2\}$ with perturbation size of 0.01. We set our target threat model to be $\ell_p, p \in \{\infty, 2\}$ perturbation size of 0.05 and compute source variation and target variation of 100 randomly sampled $h$. We sample parameters of $h$ from a standard normal distribution. We plot the linear expansion function with minimum slope in Figure 3. We find that across all settings we can find a linear expansion function that is a good fit. Additionally, we find that the slope of this linear expansion function stays consistent across changes in $\sigma$ and $d$.

We find that input dimension $n$ can influence the expansion function which matches; for example, the slope of the expansion function for source $\ell_2$ to target $\ell_\infty$ increases from $\sim 21$ to $39.09$. This matches our results in Lemma B.5 and Lemma B.7 where our computed expansion function depends on $n$.

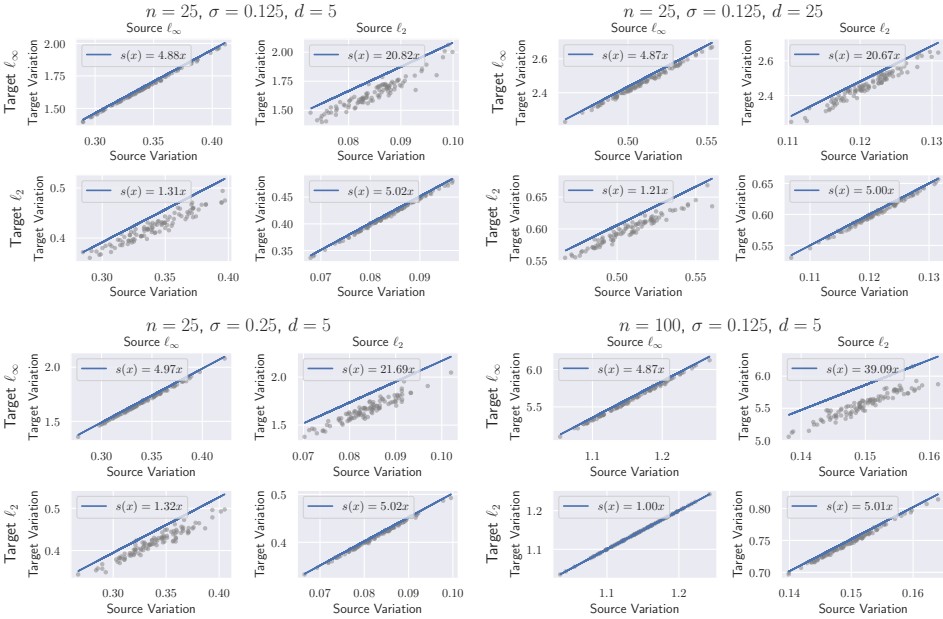

Figure 3: Plots of minimum linear expansion function $s$ shown in blue computed on 100 randomly initialized linear feature extractors across 4 different combinations of input dimension $n$, Gaussian standard deviation $\sigma$, and feature dimension $d$. Each grey point represents the variation of a single model measured across the source and target. Columns represent the source threat model ($\ell_\infty$ and $\ell_2$ with $\epsilon = 0.01$) while rows represent the target threat model ($\ell_\infty$ and $\ell_2$ with $\epsilon = 0.05$).

### D.3 Generalization curves

We plot the threat model generalization curves for varied settings of input dimension $n$, feature extractor output dimension $d$, and Gaussian standard deviation $\sigma$ in Figure 4. We find that across all settings, applying variation regularization leads to smaller generalization gap across values of $\ell_p$ radius $\epsilon$.

### D.4 Accuracies over regularization strength

We plot the accuracies corresponding to the $n = 25, \sigma = 0.125$ and $d = 5$ setting in Figure 4 in Figure 5. We note that while Figure 4 demonstrated that regularization improves decreases the size of the generalization gap, there is trade-off in accuracy which can be seen at small of $\epsilon$. However, we find that generally variation regularization improves robust accuracy on unforeseen threat models.

### D.5 Evaluations on a separate test set

We now plot generalization gap for the $n = 25, \sigma = 0.125$ and $d = 5$ setting with cross entropy loss on the target threat model measured on a separate test set of 2000 samples in Figure 6. We find that the trends we observed on the train set are consistent with those observed when evaluating on the train set shown in Figure 4.

## E  Additional Results for Logit Level AT-VR

In this section, we present additional results for AT-VR when considering the logits to be the output of the feature extractor $h$.

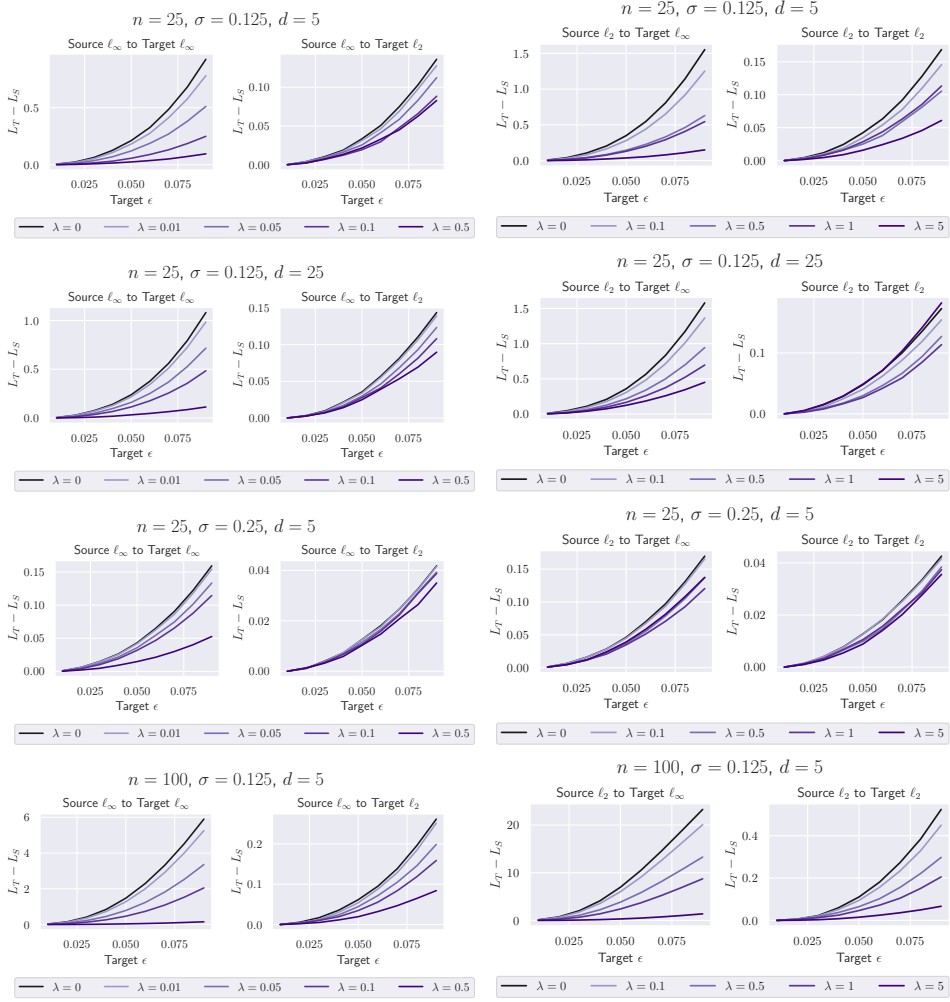

Figure 4: Threat model generalization gap of linear models on Gaussian data trained with varied variation regularization strength $\lambda$ measured on adversarial examples of generated by target $\ell_p, p = \{\infty, 2\}$ perturbations with radius $\epsilon$ at varied input dimension $n$, feature extractor output dimension $d$ and standard deviation $\sigma$. The generalization gap is measured with respect to cross entropy loss. All models are trained with source $\ell_p, p = \{\infty, 2\}$ radius of 0.01.

## E.1 Results on Additional Seeds

In Table 5 we present the mean and standard deviations for robust accuracy of CIFAR-10 ResNet-18 models over 3 trials seeded at 0, 1, and 2. We find that the improvement observed through variation regularization on unforeseen target threat models is significant for both $\ell_2$ and $\ell_\infty$ attacks for ResNet-18 CIFAR-10 models; for bolded accuracies on target threat modes (with the exception of $\ell_\infty$ source to $\ell_2$ target which we did not report as an improvement in the main text) we find that the range of the error bars do not overlap with the results for standard adversarial training ($\lambda = 0$). Additionally, we find that the trade-off with clean accuracy observed when using variation regularization is also significant.

## E.2 Expansion function on random features

In Section 5.6, we plotted the expansion function across models trained with adversarial training at various levels of variation regularization. We now visualize the expansion function for random feature extractors to investigate to what extent the learning algorithm influences expansion function. We plot the source and target variations (corresponding to the same setup as in Section 5.6) of 300 randomly

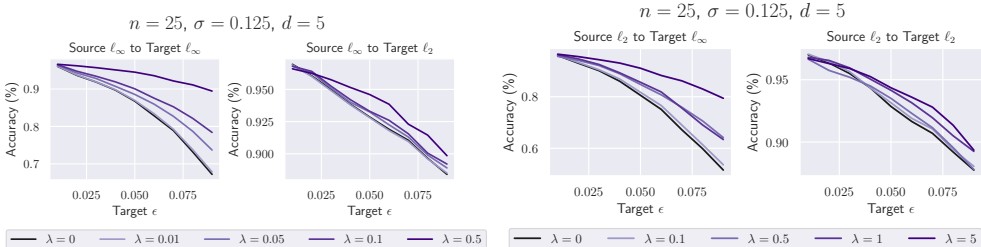

Figure 5: Accuracies of linear models on Gaussian data trained with at varied variation regularization strength $\lambda$ measured on adversarial examples of radius $\epsilon$. All models are trained with radius 0.01.

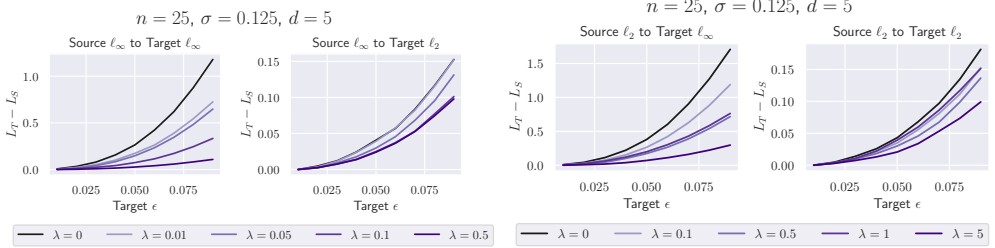

Figure 6: Cross entropy loss of linear models on Gaussian data trained with regularization strength $\lambda$ measured on adversarial examples of radius $\epsilon$. Cross entropy loss is measured with respect to a separate test set. All models are trained with source $\ell_p$ radius of 0.01.

initialized feature extractors of ResNet-18 on CIFAR-10 in Figure 7. For random initialization, we use Xavier normal initialization (Glorot & Bengio, 2010) for weights and standard normal initialization for biases.

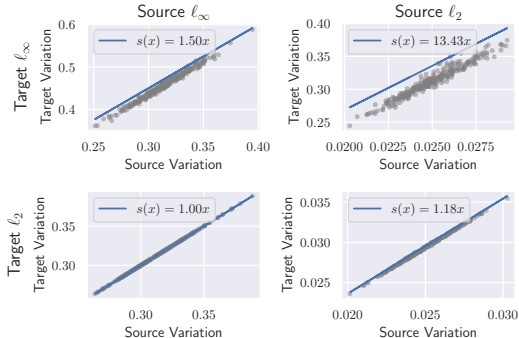

Figure 7: Plots of minimum linear expansion function $s$ shown in blue computed on 300 randomly initialized ResNet-18 models. Each grey point represents the variation measured on the source and target attack. Variation is computed on the logits. The two columns represent the source adversary ($\ell_\infty$ and $\ell_2$ respectively). The two rows represent the target adversary ($\ell_\infty$ and $\ell_2$ respectively).

We find that with randomly initialized models, we can also find a linear expansion function with small slope. In comparison to expansion functions from adversarially trained models (Figure 7), we find that using randomly initialized models leads to minimum linear expansion function $s$ with smaller slope for $\ell_\infty$ target threat model. This suggests that learning algorithm can have an impact on expansion function.

### E.3 Expansion function between $\ell_p$ and StAdv threat model

While we have shown that an expansion function with small slope exists between $\ell_2$ and $\ell_\infty$ threat models, it is unclear whether this also holds for non-$\ell_p$ threat models. However, we do observe from

| Dataset | Source | $\lambda$ | Clean acc | Source acc | Union with Source | | | | Union all |
|---|---|---|---|---|---|---|---|---|---|
| | | | | | $\ell_\infty$ $\epsilon = \frac{12}{255}$ | $\ell_2$ $\epsilon = 1$ | StAdv | Re-color | |
| CIFAR-10 | $\ell_2$ | 0 | **88.29** ± 0.51 | 67.15 ± 0.45 | 6.77 ± 0.31 | 35.40 ± 0.65 | 1.21 ± 0.46 | 66.99 ± 0.43 | 0.52 ± 0.25 |
| CIFAR-10 | $\ell_2$ | 0.25 | 87.39 ± 0.36 | 68.75 ± 0.08 | 11.56 ± 0.57 | 39.57 ± 1.08 | 10.31 ± 0.41 | 68.60 ± 0.10 | 5.93 ± 0.38 |
| CIFAR-10 | $\ell_2$ | 0.5 | 86.07 ± 0.50 | **68.78** ± 0.17 | 13.13 ± 1.33 | 41.52 ± 1.19 | 18.71 ± 2.79 | 68.59 ± 0.12 | 8.70 ± 0.50 |
| CIFAR-10 | $\ell_2$ | 0.75 | 84.54 ± 0.61 | 67.97 ± 0.15 | **14.61** ± 0.59 | **42.13** ± 0.59 | 23.24 ± 0.94 | **67.83** ± 0.12 | 10.69 ± 0.15 |
| CIFAR-10 | $\ell_2$ | 1 | 84.71 ± 1.32 | 67.39 ± 0.29 | 13.62 ± 0.40 | 41.14 ± 1.56 | **33.70** ± 5.27 | 67.29 ± 0.31 | **11.58** ± 0.37 |
| CIFAR-10 | $\ell_\infty$ | 0 | **83.01** ± 0.29 | 47.44 ± 0.08 | 27.79 ± 0.44 | 24.67 ± 0.70 | 4.17 ± 0.26 | 47.44 ± 0.08 | 2.16 ± 0.28 |
| CIFAR-10 | $\ell_\infty$ | 0.05 | 82.70 ± 0.56 | 48.38 ± 0.39 | 29.55 ± 0.43 | 25.25 ± 0.97 | 4.87 ± 0.66 | 48.38± 0.39 | 2.61 ± 0.43 |
| CIFAR-10 | $\ell_\infty$ | 0.1 | 81.79 ± 0.29 | 48.65 ± 0.25 | 29.89 ± 0.53 | 24.99 ± 0.73 | 6.33 ± 0.83 | 48.65 ± 0.25 | 3.47 ± 0.50 |
| CIFAR-10 | $\ell_\infty$ | 0.3 | 78.87 ± 0.72 | **49.16** ± 0.12 | 31.89 ± 0.13 | **24.95** ± 0.26 | 12.96 ± 0.81 | **49.16** ± 0.12 | 8.53 ± 0.70 |
| CIFAR-10 | $\ell_\infty$ | 0.5 | 74.24 ± 2.04 | 48.62 ± 0.20 | **33.07** ± 1.09 | 24.59 ± 1.32 | **19.91** ± 1.12 | 48.62 ± 0.20 | **13.35** ± 0.83 |

Table 5: Mean and standard deviation across 3 trials for robust accuracy of various models trained at different strengths of variation regularization applied on logits on various threat models. $\lambda = 0$ corresponds to standard adversarial training. Models are trained with either source threat model $\ell_\infty$ with radius $\frac{8}{255}$ or $\ell_2$ with radius 0.5. The "source acc" column reports the accuracy on the source attack. For each individual threat model, we evaluate accuracy on a union with the source threat model. The union all column reports the accuracy obtained on the union across all listed threat models.

Table 1 that AT-VR with $\ell_2$ and $\ell_\infty$ sources leads to significant improvements in robust accuracy on StAdv, which is a non-$\ell_p$ threat model, suggesting that an expansion function between these threat models exists. Using the same models for generating Figure 2, we plot the expansion function from $\ell_\infty$ and $\ell_2$ sources to StAdv ($\epsilon = 0.05$) in Figure 8. We observe that for both source threat models a linear expansion function exists to StAdv.

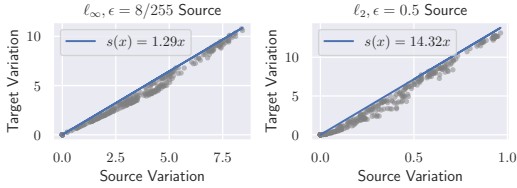

Figure 8: Expansion function from $\ell_\infty, \epsilon = 8/255$ and $\ell_2, \epsilon = 0.5$ source threat models to StAdv ($\epsilon = 0.05$) target threat model computed over 315 adversarially trained ResNet-18 models. The linear expansion function with minimum slope is plotted in blue.

We now visualize the expansion function from StAdv ($\epsilon = 0.03$) source to $\ell_\infty$ ($\epsilon = 8/255$), $\ell_2$ ($\epsilon = 0.5$), and StAdv ($\epsilon = 0.05$) target threat models. We present plots in Figure 9. Unlike our plots of expansion function for $\ell_2$ and $\ell_\infty$ source threat models, we find that for StAdv a linear expansion function is not a tight upper bound on the true trend in source vs target variation. A better model for expansion function would be piecewise linear function with 2 slopes, one for variation values near 0 and one for larger variation values since the slopes at points where source variation is closer to 0 is much larger than the slopes computed at points further from 0.

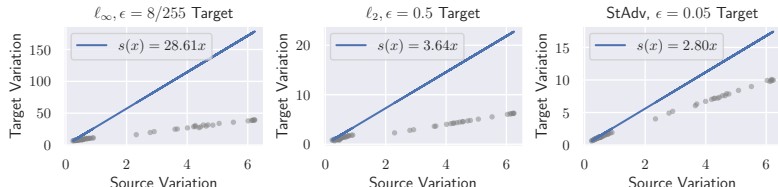

Figure 9: Expansion function from StAdv ($\epsilon = 0.03$) source threat model to $\ell_\infty$ ($\epsilon = 8/255$), $\ell_2$ ($\epsilon = 0.5$), and StAdv ($\epsilon = 0.05$) target threat models. Expansion function is computed using 60 ResNet-18 models adversarially trained on CIFAR-10 with adversarial examples generated via StAdv ($\epsilon = 0.03$). The linear expansion function with minimum slope is plotted in blue.

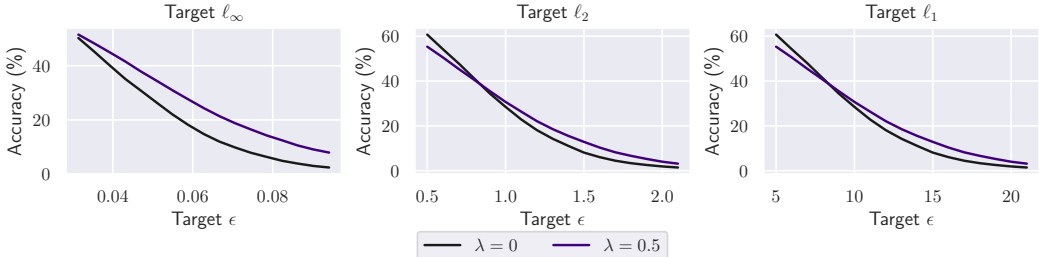

Figure 10: Robust accuracy on the CIFAR-10 test set of ResNet-18 models trained using AT-VR at varied regularization strength $\lambda$ measured on adversarial examples of generated by target $\ell_p, p = \{\infty, 2, 1\}$ perturbations with radius $\epsilon$. The generalization gap is measured with respect to cross entropy loss. Variation regularization is applied on the logits. All models are trained with source $\ell_\infty$ perturbations of radius $\frac{8}{255}$.

### E.4 Additional Results with $\ell_p$ Target Threat Models

In Figure 1 of the main text, we plotted the unforeseen generalization gap and robust accuracies for ResNet-18 models trained on CIFAR-10 with AT-VR at various perturbation size $\epsilon$ with $\ell_\infty$ source. We plot the robust accuracy across $\ell_p$ threat models for the models trained with standard adversarial training ($\lambda = 0$) and with highest variation regularization strength used ($\lambda = 0.5$) in Figure 10. We find that at large values of $\ell_p$ perturbation size, the model using variation regularization achieves higher robust accuracy than the model trained using standard adversarial training. This improvement is most clear for $\ell_\infty$ targets.

We provide corresponding plots of unforeseen generalization gap and robust accuracy on CIFAR-10 on $\ell_p$ target threat models for an $\ell_2$ source in Figure 11. Similar to trends with $\ell_\infty$ source threat model, we find that increasing the strength of variation regularization decreases the unforeseen generalization gap. We find that with an $\ell_2$ source threat model, the robust accuracy for the model trained with variation regularization also has consistently higher accuracy across target $\ell_p$ threat models compared to the model trained with standard adversarial training ($\lambda = 0$).

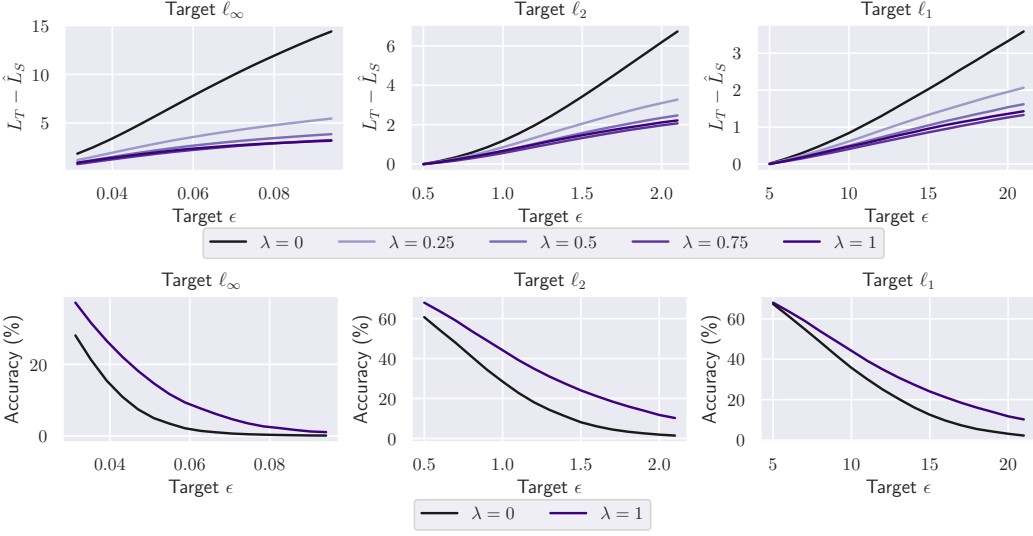

Figure 11: **Top row:** Generalization gap of on the CIFAR-10 test set for ResNet-18 models trained using AT-VR at varied regularization strength $\lambda$ measured on adversarial examples of generated by target $\ell_p, p = \{\infty, 2, 1\}$ perturbations with radius $\epsilon$. Variation regularization is applied on the logits. The generalization gap is measured with respect to cross entropy loss. All models are trained with source $\ell_2$ perturbations of radius 0.5. **Bottom row:** Corresponding robust accuracy of $\lambda = 0$ and $\lambda = 1$ models displayed in top row.

## E.5 Additional strengths of variation regularization

In Table 6 we present additional results for models trained at AT-VR at different strengths $\lambda$ of variation regularization. Overall, we find that models using variation regularization improve robustness on unforeseen attacks. Generally, we find that union accuracies are also larger with higher strengths of variation regularization.

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

Table 6: Robust accuracy of various models trained at different strengths of variation regularization applied on logits on various threat models. Models are trained with either source threat model $\ell_\infty$ with radius $\frac{8}{255}$ or $\ell_2$ with radius $0.5$. The "source acc" column reports the accuracy on the source attack. For each individual threat model, we evaluate accuracy on a union with the source threat model. The union all column reports the accuracy obtained on the union across all listed threat models.

## E.6 Full AutoAttack results on CIFAR-10

In Table 7, we report the full AutoAttack evaluation of the ResNet-18 models trained on CIFAR-10 with AT-VR with highest regularization strength displayed in in Table 13. We find that robust accuracy is consistent across attack types, suggesting that variation regularization is not obfuscating gradients.

|  |  | Source | $\ell_\infty$ | $\ell_2$ |
|  |  |  | ($\epsilon = \frac{12}{255}$) | ($\epsilon = 1$) |
| $\ell_\infty$ source $\lambda = 0.5$ | APGD-CE | 51.56 | 37.84 | 30.56 |
|  | APGD-T | 48.84 | 33.72 | 24.93 |
|  | FAB-T | 48.84 | 33.69 | 24.38 |
|  | Square | 48.84 | 33.69 | 24.38 |
| $\ell_2$ source $\lambda = 1$ | APGD-CE | 68.00 | 18.23 | 44.16 |
|  | APGD-T | 67.39 | 14.73 | 41.67 |
|  | FAB-T | 67.38 | 13.43 | 40.74 |
|  | Square | 67.38 | 13.43 | 40.74 |

Table 7: Full AutoAttack evaluations for the ResNet-18 models trained with AT-VR with variation regularization strength on $\ell_\infty$ ($\epsilon = \frac{8}{255}$) and $\ell_2$ ($\epsilon = 0.5$) source adversaries.

### E.7 Evaluations on other adversaries

In Table 8, we present additional evaluations for ResNet-18 CIFAR-10 models on additional adversaries including Wasserstein adversarial attacks, JPEG attacks, elastic attacks, and perceptual attacks. For Wasserstein adversarial attacks, we use PGD with dual projection (Wu et al., 2020b). We use $\ell_\infty$ JPEG, $\ell_1$ JPEG and elastic attacks from (Kang et al., 2019) and AlexNet LPIPS-based attacks (PPGD and LPA) from (Laidlaw et al., 2021) with perturbation size $\epsilon$ specified in Table 8. Overall, we find that using variation regularization with $\ell_p$ sources also improves performance on these additional adversaries. This suggests that an expansion function exists between variation on $\ell_p$ sources to more complicated threat models, including more perceptually-aligned threat models such as the bounded AlexNet LPIPS distance used by PPGD and LPA attacks.

| Source | $\lambda$ | Clean acc | Source acc | Union with Source |  |  |  |  |  |  |
|  |  |  |  | Wasserstein |  | $\ell_\infty$ JPEG | $\ell_1$ JPEG | Elastic | PPGD | LPA |
|  |  |  |  | $\epsilon = 0.007$ | $\epsilon = 0.01$ | $\epsilon = 0.125$ | $\epsilon = 64$ | $\epsilon = 0.25$ | $\epsilon = 0.5$ | $\epsilon = 0.5$ |
| $\ell_2$ | 0 | **88.49** | 66.65 | 45.80 | 31.75 | 48.73 | 6.42 | 13.49 | 3.65 | 0.49 |
| $\ell_2$ | 1 | 85.21 | **67.38** | **48.32** | **47.58** | **56.99** | **21.75** | **25.67** | **24.21** | **4.12** |
| $\ell_\infty$ | 0 | **82.83** | 47.47 | 26.03 | 16.62 | 34.45 | 2.32 | 25.23 | 2.44 | 0.28 |
| $\ell_\infty$ | 0.5 | 72.91 | **48.84** | **29.50** | **20.89** | **36.63** | **7.25** | **29.62** | **4.96** | **2.18** |

Table 8: Robust accuracy of ResNet-18 models on CIFAR-10 evaluated on additional adversaries including Wasserstein adversaries (Wu et al., 2020b), JPEG compression adversaries and elastic adversary (Kang et al., 2019), and AlexNet LPIPS perceptual adversaries (Laidlaw et al., 2021). Perturbation size $\epsilon$ is specified for each threat model. Accuracies on each target adversary reported are given with a union computed on the source.

### E.8 Comparison to training with all attacks

In this section, we compare to the MAX adversarial training approach from Tramèr & Boneh (2019) which trains directly on all target threat models of interest. Since MAX training uses information about the target threat model for training while our approach does not, the union accuracies achieved via MAX training should be viewed as an upper bound on performance. For models using MAX training, we train for a total of 100 epochs and evaluate performance on the model saved at the final epoch. We compare robust accuracies for MAX training to robust accuracies of models trained with variation regularization from Table 1. For fair comparison, we report robust accuracies of models trained with variation regularization measured through evaluating on the target threat model (instead of the union of target with source threat model as in Table 1. We provide results in Table 9. We note that in general, training with the union of all attacks achieves more balanced accuracies across threat models. The only exception is with WRN-28-10 for which MAX training achieves lower union accuracy and StAdv accuracy; however, this may be due to using only 100 epochs of training.

### E.9 Combining variation regularization with TRADES

In the main paper, we provided experiments with variation regularization combined with PGD adversarial training (Madry et al., 2018) on $\ell_\infty$ and $\ell_2$ source threat models. We note that variation regularization is not exclusive to PGD adversarial training and can be applied in conjunction with other adversarial training based techniques including TRADES (Zhang et al., 2019). We present results

| Dataset | Architecture | Source | $\lambda$ | Clean acc | $\ell_\infty$ $\epsilon = \frac{12}{255}$ | $\ell_2$ $\epsilon = 1$ | StAdv | Re-color | Union all |
|---|---|---|---|---|---|---|---|---|---|
| CIFAR-10 | ResNet-18 | MAX | 0 | 77.81 | 41.76 | 36.74 | 38.90 | 63.99 | 27.23 |
| CIFAR-10 | ResNet-18 | $\ell_2$ | 1 | 85.21 | 13.39 | 40.79 | 32.15 | 60.07 | 11.27 |
| CIFAR-10 | ResNet-18 | $\ell_\infty$ | 0.5 | 72.91 | 33.66 | 24.34 | 16.09 | 47.38 | 10.58 |
| CIFAR-10 | WRN-28-10 | MAX | 0 | 80.98 | 29.78 | 38.25 | 9.83 | 57.60 | 4.98 |
| CIFAR-10 | WRN-28-10 | $\ell_\infty$ | 0.7 | 72.73 | 35.08 | 22.35 | 23.15 | 49.01 | 13.26 |
| CIFAR-10 | VGG-16 | MAX | 0 | 67.26 | 29.00 | 41.48 | 42.98 | 59.11 | 26.12 |
| CIFAR-10 | VGG-16 | $\ell_\infty$ | 0.1 | 77.80 | 28.38 | 32.08 | 11.96 | 62.24 | 5.38 |
| ImageNette | ResNet-18 | MAX | 0 | 69.32 | 38.57 | 64.36 | 57.04 | 59.34 | 37.25 |
| ImageNette | ResNet-18 | $\ell_2$ | 1 | 85.22 | 9.50 | 80.43 | 16.25 | 44.99 | 5.83 |
| ImageNette | ResNet-18 | $\ell_\infty$ | 0.1 | 78.01 | 35.54 | 72.94 | 51.75 | 64.15 | 31.03 |
| CIFAR-100 | ResNet-18 | MAX | 0 | 52.06 | 13.29 | 19.35 | 8.01 | 26.42 | 5.04 |
| CIFAR-100 | ResNet-18 | $\ell_2$ | 0.75 | 51.53 | 11.46 | 25.64 | 4.17 | 18.98 | 2.26 |
| CIFAR-100 | ResNet-18 | $\ell_\infty$ | 0.2 | 48.97 | 16.51 | 15.81 | 4.70 | 24.01 | 2.59 |

Table 9: Robust accuracy of various models trained at different strengths of VR applied on logits on various threat models. Source of MAX represents the accuracies obtained by directly training on all target threat models. The union all column reports the accuracy on the union across all listed threat models.

for TRADES-VR with TRADES hyperparameter $\beta = 6.0$ in Table 10. Interestingly, we find that compared to PGD adversarial training results in Table 1, TRADES generally has better performance on larger $\ell_\infty$ and $\ell_2$ threat models (on par with AT-VR with PGD). We find that across architectures and datasets applying variation regularization over TRADES adversarial training generally improves robustness on unforeseen $\ell_\infty$ and StAdv threat models, but trades off clean accuracy, source accuracy, and unforeseen $\ell_2$ target accuracy. Despite this trade-off, we find that TRADES-VR consistently improves on the accuracy across the union of all threat models in comparison to standard TRADES.

| Dataset | Architecture | Source | $\lambda$ | Clean acc | Source acc | Union with Source $\ell_\infty$ $\epsilon = \frac{12}{255}$ | $\ell_2$ $\epsilon = 1$ | StAdv | Re-color | Union all |
|---|---|---|---|---|---|---|---|---|---|---|
| CIFAR-10 | ResNet-18 | $\ell_2$ | 0 | **86.79** | 68.99 | 13.13 | **43.58** | 1.89 | **68.85** | 1.00 |
| CIFAR-10 | ResNet-18 | $\ell_2$ | 2 | 79.81 | 64.84 | **16.04** | 42.46 | **42.05** | 64.76 | **14.67** |
| CIFAR-10 | ResNet-18 | $\ell_\infty$ | 0 | **82.67** | 49.15 | 31.00 | **28.04** | 5.32 | **49.15** | 3.64 |
| CIFAR-10 | ResNet-18 | $\ell_\infty$ | 0.5 | 79.11 | 48.98 | **32.08** | 26.54 | **16.21** | 48.98 | **11.78** |
| CIFAR-10 | WRN-28-10 | $\ell_\infty$ | 0 | **84.73** | 52.09 | 32.74 | **24.68** | 4.54 | **52.09** | 2.98 |
| CIFAR-10 | WRN-28-10 | $\ell_\infty$ | 1 | 75.99 | 50.09 | **33.96** | 21.76 | **25.09** | 50.09 | **15.08** |
| ImageNette | ResNet-18 | $\ell_2$ | 0 | **88.66** | 85.55 | 0.08 | **80.71** | 1.94 | 72.74 | 0.03 |
| ImageNette | ResNet-18 | $\ell_2$ | 2 | 83.80 | 82.27 | **14.37** | 80.10 | **27.08** | **75.03** | **11.75** |
| ImageNette | ResNet-18 | $\ell_\infty$ | 0 | **78.32** | 50.62 | 35.54 | **50.62** | 44.03 | **50.62** | 33.32 |
| ImageNette | ResNet-18 | $\ell_\infty$ | 0.2 | 73.83 | 48.79 | **36.15** | 48.79 | **44.97** | 48.79 | **34.62** |
| CIFAR-100 | ResNet-18 | $\ell_2$ | 0 | **58.71** | 37.79 | 6.73 | 21.40 | 2.72 | **36.74** | 1.21 |
| CIFAR-100 | ResNet-18 | $\ell_2$ | 1 | 53.44 | 37.71 | **10.09** | **24.55** | **4.05** | 36.59 | **2.40** |
| CIFAR-100 | ResNet-18 | $\ell_\infty$ | 0 | **53.80** | 23.02 | 13.77 | **15.21** | **4.94** | 22.99 | 3.33 |
| CIFAR-100 | ResNet-18 | $\ell_\infty$ | 0.5 | 51.16 | **24.39** | **15.93** | 14.33 | 4.87 | **24.34** | **3.40** |

Table 10: Robust accuracy of various models trained with TRADES-VR at different strengths of variation regularization applied on logits on various threat models. Models are trained with either source threat model $\ell_\infty$ with radius $\frac{8}{255}$ or $\ell_2$ with radius 0.5. The "source acc" column reports the accuracy on the source attack. For each individual threat model, we evaluate accuracy on a union with the source threat model. The union all column reports the accuracy obtained on the union across all listed threat models.

## E.10 Combining variation regularization with other sources

To demonstrate that variation regularization can be applied with source threat models outside of $\ell_p$ balls, we evaluate the performance of AT-VR with other sources including StAdv and Recolor.

### E.10.1 Computing variation with StAdv and Recolor sources

In StAdv (Xiao et al., 2018), adversarial examples are generated by optimizing for a per pixel flow field $f$, where $f_i$ corresponds to the displacement vector of the $i$th pixel of the image. This flow field

| Source | $\lambda$ | Clean acc | Source acc | Union with Source | | | | Union all |
|---|---|---|---|---|---|---|---|---|
| | | | | $\ell_\infty$ $\epsilon = \frac{4}{255}$ | $\ell_2$ $\epsilon = 0.5$ | StAdv | Re-color | |
| StAdv | 0 | **86.94** | 54.04 | 3.57 | 5.63 | 12.62 | 3.27 | 0.96 |
| StAdv | 0.5 | 83.88 | 60.11 | 3.29 | 5.66 | 24.64 | 8.60 | 2.39 |
| StAdv | 1 | 81.24 | **62.78** | **5.83** | **9.97** | **31.09** | **13.10** | **4.44** |
| Recolor | 0 | **94.88** | 39.11 | 5.38 | 3.07 | 0.00 | 21.75 | 0.00 |
| Recolor | 0.5 | 94.18 | 71.36 | 25.39 | 19.22 | 0.06 | 64.39 | 0.03 |
| Recolor | 1 | 94.13 | **72.92** | **26.81** | **20.03** | **0.20** | **66.02** | **0.17** |

Table 11: Robust accuracy of ResNet-18 models trained using AT-VR with StAdv and Recolor source threat models with variation regularization applied on logits on various threat models. During training we use 0.03 and 0.04 as the perturbation bounds for StAdv and Recolor respectively. During testing we use 0.05 for StAdv and 0.06 for Recolor. The "source acc" column reports the accuracy on the source attack. For each individual threat model, we evaluate accuracy on a union with the source threat model. The union all column reports the accuracy obtained on the union across all listed threat models.

is obtained by solving:

$$\arg\min_f \ell_{\text{adv}}(x, f) + \tau \ell_{\text{flow}}(f) \tag{3}$$

where $\ell_{\text{adv}}$ is the CW objective (Carlini & Wagner, 2017) and $\ell_{flow}$ is a regularization term minimizing spatial transformation distance to ensure that the perturbation is imperceptible. $\tau$ controls the strength of this regularization.

We adapt this objective to compute variation, replacing $\ell_{\text{adv}}$ with the variation objective. Rather than solving for a single flow field, we solve

$$\mathcal{V}(h, \text{StAdv}_\tau) = \max_{f_1, f_2} ||h(f_1(x)) - h(f_2(x))||_2 - \tau(\ell_{\text{flow}}(f_1) + \ell_{\text{flow}}(f_2))$$

Here $f_1(x)$ and $f_2(x)$ denote the perturbed input after applying $f_1$ and $f_2$ respectively and $h$ denotes our feature extractor. The second term ensures that both $f_1$ and $f_2$ have small spatial transformation distance. We solve the optimization problem using PGD.

In Recolor attacks (Laidlaw & Feizi, 2019), the objective function takes the same form as Equation 3 where $f$ is now a color perturbation function. We optimize for adversarial examples for Recolor in the same way as for StAdv, but incorporate an additional clipping step to ensure that perturbations of each color are within the specified bounds.

### E.10.2   Experimental setup details for StAdv and Recolor sources

We train ResNet-18 models using AT-VR with StAdv and Recolor sources. For these models, we train with source perturbation bound of 0.03 for StAdv and 0.04 for Recolor attacks. We use 10 iterations for both StAdv and Recolor during training. We train for a maximum of 100 epochs and evaluate on the model saved at epoch achieving the highest source accuracy. For evaluation, we use StAdv perturbation bound of 0.05 and Recolor perturbation bound of 0.06 and use 100 iterations to generate adversarial examples for both attacks. Additionally, we evaluate on $\ell_\infty$ and $\ell_2$ attacks with radius $\frac{4}{255}$ and 0.5 respectively using AutoAttack.

### E.10.3   Experimental Results for StAdv and Recolor sources

We report results in Table 11. Similar to trends observed with $\ell_p$ sources, we find that using variation regularization improves unforeseen robustness when using StAdv and Recolor sources. For example, with StAdv source, using variation regularization with $\lambda = 1$ increases robust accuracy on unforeseen $\ell_2$ attacks from 3.57% to 5.83%, and with Recolor source, using variation regularization with $\lambda = 1$ increases robust accuracy on unforeseen $\ell_2$ attacks from 3.07% to 20.03%. The largest increase is for attacks of the same perturbation type but larger radius; for example, for StAdv source ($\epsilon = 0.03$) on unforeseen StAdv target ($\epsilon = 0.05$), the robust accuracy increases from 12.62% to 31.09%. Similarly, for Recolor source ($\epsilon = 0.04$) on unforeseen Recolor target ($\epsilon = 0.06$), the robust accuracy increases from 21.75% to 66.02%. Interestingly, we also find that using variation regularization with StAdv and Recolor sources leads to a significant increase in source accuracy as well. For StAdv, source accuracy increases from 54.04% without variation regularization to 62.78% with variation regularization at

$\lambda = 1$. For Recolor source, this increase is even larger; source accuracy increases from 39.11% without variation regularization to 72.92% with variation regularization.

Further inspecting the source accuracy of the models trained Recolor source, we find that without variation regularization, the model overfits to the 10-iteration attack used during training. When evaluated with 10-iteration Recolor, the standard adversarial training ($\lambda = 0$) model achieves 86.62% robust accuracy, but when 100 iterations of the attack is used during testing, the model's accuracy drops to 39.11%. Interestingly, variation regularization helps prevent adversarial training from overfitting to the 10-iteration attack, causing the resulting source accuracy on the models with variation regularization to be significantly higher.

### E.11  Computational complexity of AT-VR

One limitation of AT-VR is that it can be 3x as expensive compared to adversarial training. This is because the computation for variation also requires gradient based optimization. We note that this computational expense occurs when we use the same number of iterations of PGD for variation computation as standard adversarial training. In this section, we study whether we can use fewer iterations of PGD for generating the adversarial example and computing variation. We present our results for training ResNet-18 on CIFAR-10 with source threat model $\ell_2, \epsilon = 0.5$ using AT-VR and standard AT ($\lambda = 0$) in Table 12. We find that even with a single iteration, AT-VR is able to significantly improve union accuracy over standard AT. For $\ell_\infty$ adversarial training, we find that a

| $\lambda$ | PGD iters | Clean acc | Source acc | Union with Source | | | | Union all |
| --- | --- | --- | --- | --- | --- | --- | --- | --- |
| | | | | $\ell_\infty$ ($\epsilon = \frac{12}{255}$) | $\ell_2$ ($\epsilon = 1$) | StAdv | Recolor | |
| 0 | 1 | **89.00** | 66.53 | 5.54 | 31.55 | 0.26 | 33.43 | 0.05 |
| 0 | 3 | 88.72 | **67.58** | 7.07 | 35.47 | 0.55 | 36.41 | 0.18 |
| 0 | 10 | 88.49 | 66.65 | 6.44 | 34.72 | 0.76 | **66.52** | 0.33 |
| 1 | 1 | 86.88 | 67.00 | **11.52** | **37.24** | **38.34** | 64.44 | **10.09** |

Table 12: Robust accuracy of ResNet-18 models trained on CIFAR-10 using AT-VR with a single PGD iteration ($\lambda = 1$, PGD iters=1) in comparison to standard adversarial training ($\lambda = 0$) with various numbers of PGD iterations.

single iteration of training does not work well due to the poor performance of adversarial training with FGSM.

## F  Additional Results for Feature Level AT-VR

In the main paper, we provide results for AT-VR with variation regularization applied at the layer of the logits. In terms of our theory, this would correspond to considering the identity function to be the top level classifier. In this section, we consider the top level classifier to be all fully connected layers at the end of the NN architectures used and evaluate variation regularization applied at the input into the fully connected classifier.

### F.1  Expansion function for variation on features

In Figure 12, we plot the minimum linear expansion function computed on 315 adversarially trained feature extractors (analogous to Figure 2 in the main text). Additionally, we plot the minimum linear expansion function on 300 randomly initialized feature extractors. For random initialization, we use Xavier normal initialization (Glorot & Bengio, 2010) for weights and standard normal initialization for biases. We find that we can find a linear expansion function with small slope across $\ell_\infty$ and $\ell_2$ source and target pairs. In comparison to expansion function for variation computed at the logits, we find that the slope of the expansion function $s$ found is similar.

### F.2  Additional results with $\ell_p$ target threat models

We present the unforeseen generalization gap for ResNet-18 models on CIFAR-10 trained with source $\ell_\infty$ threat model with radius $\frac{8}{255}$ at various strengths of variation regularization and the corresponding

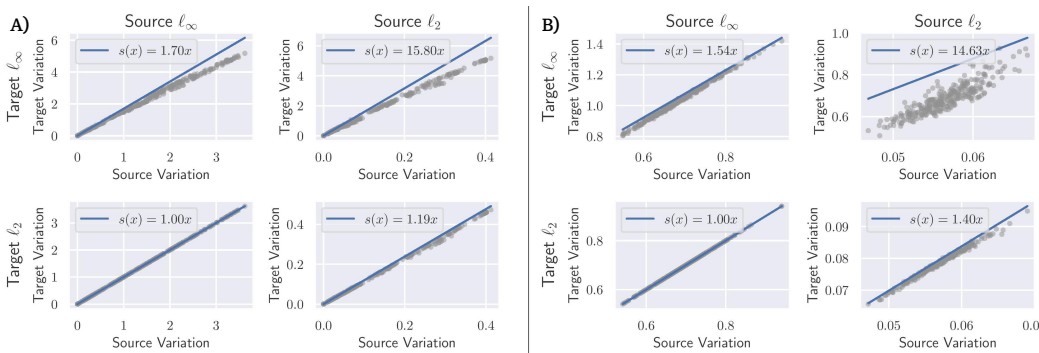

Figure 12: Plots of minimum linear expansion function $s$ shown in blue computed on A) 315 adversarially trained feature extractors and B) 300 randomly initialized feature extractors. Each grey point represents the variation measured on the source and target attack. The two columns represent the source adversary ($\ell_\infty$ and $\ell_2$ respectively). The two rows represent the target adversary ($\ell_\infty$ and $\ell_2$ respectively).

robust accuracy of the model trained with standard AT ($\lambda = 0$) and highest variation regularization ($\lambda = 2$) in Figure 13. We find at large values of unforeseen $\epsilon$, the model trained with variation regularization achieve both smaller unforeseen generalization gap and higher robust accuracy.

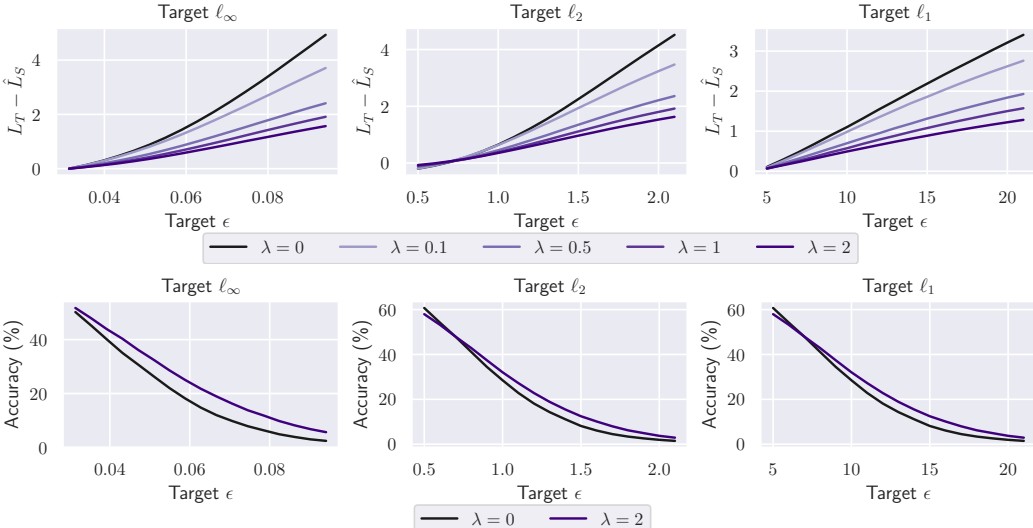

Figure 13: Top row: Unforeseen generalization gap of on the CIFAR-10 test set for ResNet-18 models trained using AT-VR at varied regularization strength $\lambda$ measured on adversarial examples of generated by target $\ell_p, p = \{\infty, 2, 1\}$ perturbations with radius $\epsilon$. The generalization gap is measured with respect to cross entropy loss. All models are trained with source $\ell_\infty$ perturbations of radius $\frac{8}{255}$. Bottom row: Corresponding robust accuracy of $\lambda = 0$ and $\lambda = 2$ models displayed in top row.

We repeat experiments with ResNet-18 models on CIFAR-10 trained with source $\ell_2$ threat model with radius of 0.5. We report the measured unforeseen generalization gap to $\ell_\infty, \ell_2$, and $\ell_1$ target threat models at different radii (measured via cross entropy loss on adversarial examples generated with APGD) along with corresponding robust accuracy of the no regularization and maximum regularization strength models in Figure 14. We find that trends observed when the source threat model was $\ell_\infty$ are consistent with the trends for $\ell_2$ source threat model: increasing the strength of variation regularization decreases the size of the unforeseen generalization gap and increases robust accuracy across various $\ell_p$ target threat models.

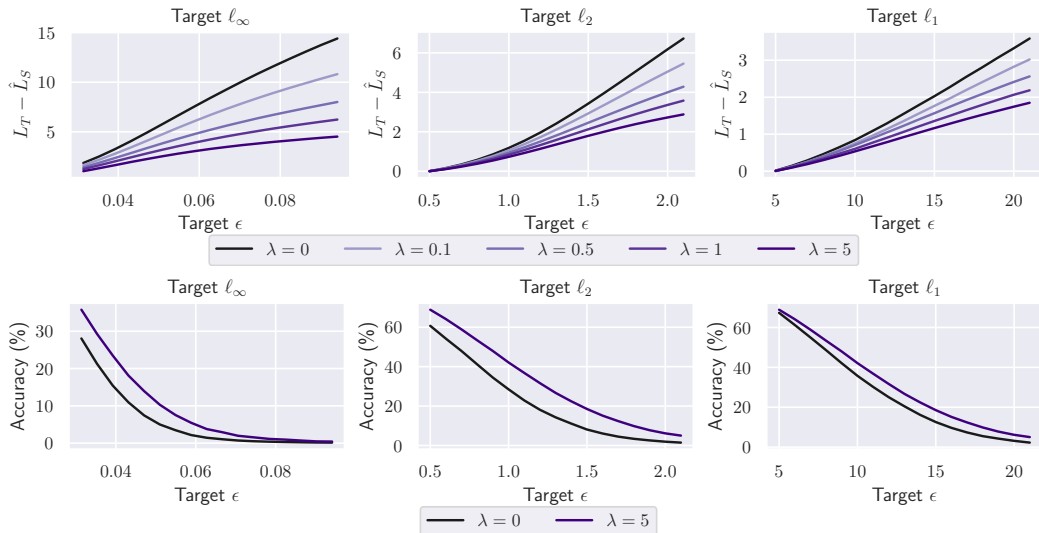

Figure 14: Top row: Unforeseen generalization gap of on the CIFAR-10 test set for ResNet-18 models trained using AT-VR at varied regularization strength $\lambda$ measured on adversarial examples of generated by target $\ell_p, p = \{\infty, 2, 1\}$ perturbations with radius $\epsilon$. The generalization gap is measured with respect to cross entropy loss. All models are trained with source $\ell_2$ perturbations of radius 0.5. Bottom row: Corresponding robust accuracy of $\lambda = 0$ and $\lambda = 5$ models displayed in top row.

## F.3 Robust accuracies with feature level AT-VR

We repeat experiments corresponding to Table 1 in the main paper for models trained with feature level AT-VR in Table 13. We observe similar trends with variation regularization applied at the features instead of logits: variation regularization improves unforeseen accuracy and generally improves source accuracy, but trades off performance on clean images.

| | | | | | | Union with Source | | | | |
|---|---|---|---|---|---|---|---|---|---|---|
| Dataset | Architecture | Source | $\lambda$ | Clean acc | Source acc | $\ell_\infty$ ($\epsilon = \frac{12}{255}$) | $\ell_2$ ($\epsilon = 1$) | StAdv | Recolor | Union all |
| CIFAR-10 | ResNet-18 | $\ell_2$ | 0 | **88.49** | 66.65 | 6.44 | 34.72 | 0.76 | 66.52 | 0.33 |
| CIFAR-10 | ResNet-18 | $\ell_2$ | 2.0 | 86.87 | **68.24** | 12.66 | 41.05 | 10.01 | **68.06** | 6.16 |
| CIFAR-10 | ResNet-18 | $\ell_2$ | 5.0 | 84.75 | 66.93 | **13.29** | **40.71** | **29.20** | 66.84 | **10.83** |
| CIFAR-10 | ResNet-18 | $\ell_\infty$ | 0 | **82.83** | 47.47 | 28.09 | 24.94 | 4.38 | 47.47 | 2.48 |
| CIFAR-10 | ResNet-18 | $\ell_\infty$ | 1.0 | 79.72 | **49.43** | **32.57** | 26.64 | 11.38 | **49.43** | 7.09 |
| CIFAR-10 | ResNet-18 | $\ell_\infty$ | 2.0 | 75.58 | 48.35 | 32.19 | **26.89** | **16.56** | 48.35 | **11.44** |
| CIFAR-10 | WRN-28-10 | $\ell_\infty$ | 0 | **85.93** | 49.86 | 28.73 | 20.89 | 2.28 | 49.86 | 1.10 |
| CIFAR-10 | WRN-28-10 | $\ell_\infty$ | 0.5 | 85.86 | 50.13 | 30.04 | 21.62 | 5.36 | 50.13 | 4.14 |
| CIFAR-10 | WRN-28-10 | $\ell_\infty$ | 1 | 84.27 | **51.01** | **31.47** | **22.86** | **9.71** | **51.01** | **7.59** |
| CIFAR-10 | VGG-16 | $\ell_\infty$ | 0 | **79.67** | 44.36 | 26.14 | 30.82 | 7.31 | 44.36 | 4.35 |
| CIFAR-10 | VGG-16 | $\ell_\infty$ | 0.01 | 76.38 | **44.87** | **27.35** | **32.59** | 9.14 | **44.87** | 5.69 |
| CIFAR-10 | VGG-16 | $\ell_\infty$ | 0.05 | 72.27 | 42.14 | 26.80 | 32.41 | **12.18** | 42.14 | **8.02** |
| ImageNette | ResNet-18 | $\ell_2$ | 0 | **88.94** | **84.99** | 0.00 | 79.08 | 1.27 | 72.15 | 0.00 |
| ImageNette | ResNet-18 | $\ell_2$ | 1.0 | 86.29 | 83.62 | 2.55 | **80.20** | 8.66 | 73.25 | 1.45 |
| ImageNette | ResNet-18 | $\ell_2$ | 5.0 | 83.06 | 80.89 | **10.11** | 78.60 | **22.98** | **74.22** | **7.75** |
| ImageNette | ResNet-18 | $\ell_\infty$ | 0 | **80.56** | 49.63 | 32.38 | 49.63 | 34.27 | 49.63 | 25.68 |
| ImageNette | ResNet-18 | $\ell_\infty$ | 0.05 | 79.06 | **50.47** | 34.06 | **50.47** | 37.40 | **50.47** | 28.89 |
| ImageNette | ResNet-18 | $\ell_\infty$ | 0.1 | 78.09 | 50.01 | **34.11** | 50.01 | **38.32** | 50.01 | **29.30** |
| CIFAR-100 | ResNet-18 | $\ell_2$ | 0 | **60.92** | 36.01 | 3.98 | 16.90 | 1.80 | 34.87 | 0.40 |
| CIFAR-100 | ResNet-18 | $\ell_2$ | 1 | 56.37 | **38.66** | 8.65 | **23.41** | 4.81 | **37.52** | 2.10 |
| CIFAR-100 | ResNet-18 | $\ell_2$ | 2 | 52.73 | 36.15 | **8.76** | 22.33 | **7.46** | 35.28 | **3.14** |
| CIFAR-100 | ResNet-18 | $\ell_\infty$ | 0 | **54.94** | 22.74 | 12.61 | 14.40 | 3.99 | 22.71 | 2.42 |
| CIFAR-100 | ResNet-18 | $\ell_\infty$ | 0.1 | 54.21 | 23.52 | 13.61 | 15.10 | 4.10 | 23.48 | 2.54 |
| CIFAR-100 | ResNet-18 | $\ell_\infty$ | 0.5 | 49.29 | **24.66** | **16.02** | **15.62** | 5.74 | 24.58 | **3.70** |

Table 13: Robust accuracy of various models trained at different strengths of variation regularization on various threat models. Models are trained with either source threat model $\ell_\infty$ with radius $\frac{8}{255}$ or $\ell_2$ with radius 0.5. The "source acc" column reports the accuracy on the source attack. For each individual threat model, we evaluate accuracy on a union with the source threat model. The union all column reports the accuracy obtained on the union across all listed threat models.