# OpenReview forum: "Formulating Robustness Against Unforeseen Attacks"
_NeurIPS.cc/2022/Conference — NeurIPS 2022 Accept_

### Official Review · Reviewer_9qwD · 2022-06-19

**Rating:** 6
**Confidence:** 4
**Soundness:** 2 fair
**Presentation:** 3 good
**Contribution:** 3 good

**Summary:**

This paper studies a scenario where the threat model assumed by the defense during training (i.e. source threat model S) is different from the threat model used during test time (i.e. target threat model T). It assumes that the target threat model is unknown and is stronger than the source threat model (i.e. the target threat model T is a strict superset of the source threat model S). It formally defines the problem of learning and generalization with an unforeseen adversary. It derives a generalization bound that relates the generalization gap between source and target threat models to variation of the feature extractor, which measures the expected maximum difference between extracted features across a given threat model. Based on the generalization bound, it proposes adversarial training with variation regularization (AT-VR), which reduces the variation of the feature extractor across the source threat model during training. The experimental results demonstrate that AT-VR can lead to improved generalization to unforeseen attacks during test-time compared to standard adversarial training.

**Questions:**

1. Could Theorem 4.7 be extended to non-linear cases?

2. Could the authors add more baselines in Table 1?

3. How to select the hyper-parameter $\lambda$?

4. Could the authors provide some empirical evidence to support the hypothesis stated in Line 380-381?

**Limitations:**

The limitations and potential negative societal impact are properly addressed.

**Strengths And Weaknesses:**

I think this paper has the following strengths:

1. It studies an important problem: generalizing the robustness to a stronger unknown target threat model during test time. Existing works usually assume that the threat model during test time is the same as the threat model during training time, which is not realistic. This paper makes a key contribution by formally defining the problem of learning and generalization with an unforeseen adversary.

2. Under the framework for generalizability, it derives a generalization bound for generalization across threat models. The bound relates the generalization gap to a quantity defined as the variation. It shows that under certain conditions, this upper bound can be decreased while only using information about the source threat model. Based on the generalization bound, it proposes a learning algorithm AT-VR for unforeseen robustness. Empirically, it demonstrates that this technique can lead to improved robustness on unforeseen attacks across datasets such as CIFAR-10, CIFAR-100, and ImageNette over standard adversarial training.

3. It is well-written and the related works are properly discussed.

However, I think this paper has the following weaknesses:

1. The source variation-based threat model generalization bound (corollary 4.4) relies on an assumption: an expansion function s from the source threat model S to the target threat model T exists. This assumption may be too strong. Although Theorem 4.7 shows that in the case of a linear feature extractor, a linear expansion function exists for any data distribution from a source $\ell_p$ adversary to a union of $\ell_p$ adversaries, it is unclear if such expansion functions exist for neural networks. The linear feature extractor with bounded condition number is too simple. It can consider a more complex case (e.g. two-layer neural networks) and check if the expansion function exists in such a case.

2. In Table 1, it only considers one baseline standard adversarial training. I think it should add more baselines. For example, it can compare the proposed method AT-VR to the Perceptual Adversarial Training (PAT) (Laidlaw et al. 2021). Also, it can add the baseline of adversarial training against the target threat model as an "oracle" baseline.

3. It doesn't discuss how to select the hyper-parameter $\lambda$. Note that the best hyper-parameters should not be selected using the test data.

4. In Line 380-381, it states that "we hypothesize that this property also appears for ResNet-18 models because neural networks are piecewise linear". I think it should provide some empirical evidence to support this hypothesis.

---

> ### Author Response · Authors · 2022-08-02
> **Author response (pt 1)**
>
> Thank you for the feedback and questions.  We are encouraged that you find the problem of robustness to unforeseen attacks important and find our formulation of learning and generalization in this setting a key contribution.  We provide our responses to your comments and questions below:
>
> > The source variation-based threat model generalization bound (Corollary 4.4) relies on an assumption: an expansion function s from the source threat model S to the target threat model T exists. This assumption may be too strong… it is unclear if such expansion functions exist for neural networks.
>
> We note that a relationship between the source threat model and target threat model must exist otherwise we cannot expect loss on the source threat model to be reflective of loss on the target threat model.  An example of this is when the source threat model is a small $\ell_2$ ball and the target threat model is the space of unbounded perturbations.  We would not expect robustness on the source threat model to mean anything about robustness on the target threat model.
>
> Empirically, we find that for neural networks the existence of an expansion function is not an unreasonable assumption.  In Figure 2, we plot the expansion function between $\ell_{\infty}$ and $\ell_2$ threat models for adversarially trained ResNet-18 models.  We find that for these models expansion functions still exist between $\ell_{\infty}$ and $\ell_2$ threat models.  In our response to Reviewer PVxo, we include additional values for expansion function slopes to and from StAdv to further demonstrate that even between non-$\ell_p$ and $\ell_p$ norms, an expansion function exists.
>
> > Could the authors provide some empirical evidence to support the hypothesis stated in Line 380-381?
>
> The statement “we hypothesize that this property also appears for ResNet-18 models because neural networks are piecewise linear” refers to the results in Figure 2, where we see that there exists a linear expansion function with small slope between $\ell_p$ threat models.  This interestingly matches the statement from Theorem 4.7 which holds only for linear models, which is why we made this hypothesis.
>
> > Could the authors add more baselines in Table 1?
>
> We would like to point out that VR can be combined with robust training techniques such as TRADES and PAT. In Appendix F.1.7, we include a comparison of TRADES-VR to TRADES.  In Appendix F.1.8 Table 8, we present results for combining VR with PAT and compare it to PAT.  We find that variation regularization improves robust accuracy on the union of l2, linf, recolor, and StAdv attacks over PAT, while also significantly increasing robustness on the LPA attack (PAT achieves 9.8% robust accuracy with LPA attacks, while PAT-VR achieves 30.8% robust accuracy).  We agree with the reviewer that this is an important comparison and will move this table into the main text since we are allowed an additional page in the camera-ready submission.
>
> Following reviewer advice, we present results for adversarial training with all target threat models (using MAX technique from Tramer et al. 2019) for ResNet-18 models trained on CIFAR-10 below:
>
> | Clean Acc | Linf ($\epsilon=\frac{12}{255}$) Acc | L2 ($\epsilon=1$) Acc | StAdv Acc | Recolor Acc | Union Acc |
> |-----------|--------------------------------------|-----------------------|-----------|-------------|-----------|
> | 77.81     | 41.76                                | 36.74                 | 38.90     | 63.99       | 27.23     |
>
> We are working on obtaining the robust accuracies for other architectures and datasets present in Table 1 and will incorporate these results into the Appendix of the camera ready submission.  We also note that the union accuracies from MAX training should be viewed as an upper bound on performance since it uses full knowledge of the target threat model while our algorithm does not use knowledge of the target threat model.
>
> Tramer, Florian, and Dan Boneh. "Adversarial training and robustness for multiple perturbations." Advances in Neural Information Processing Systems 32 (2019).

---

> > ### Author Response · Authors · 2022-08-02
> > **Author response (pt 2)**
> >
> > > How to select the hyper-parameter?
> >
> > For the results in Table 1, we took the results for the $\lambda$ value with highest union accuracy in Table 3 in the Appendix.  Table 3 shows the robust accuracies at various values of $\lambda$ and we find that generally increasing $\lambda$ increases robustness on the source threat model and on the union of unforeseen threat models (some error bars for the ResNet-18 CIFAR-10 models are shown in Appendix Table 2).  Generally any $\lambda > 0$ improves over adversarial training.  In practice, since the learner cannot measure accuracy on unforeseen threat models, the value of lambda should be chosen based on how much clean accuracy the learner is willing to trade off.  For this, the learner can use a validation set to estimate the clean accuracy at different settings of $\lambda$.
> >
> > In our experiments, the remaining hyperparameters such as learning rate for adversarial training are taken from the setup used in Gowal et al. 2020.
> >
> > Gowal, Sven, et al. "Uncovering the limits of adversarial training against norm-bounded adversarial examples." arXiv preprint arXiv:2010.03593 (2020).

---

> > ### Comment · Reviewer_9qwD · 2022-08-06
> > **Response to the authors**
> >
> > Thanks to the authors for addressing my concerns. I think most of my concerns have been addressed. Thus, I increase my overall score from 5 to 6.

---

### Official Review · Reviewer_UuTG · 2022-07-11

**Rating:** 5
**Confidence:** 3
**Soundness:** 3 good
**Presentation:** 2 fair
**Contribution:** 2 fair

**Summary:**

This paper studies the model robustness extension to unforeseen perturbations. The main contribution of the paper is a generalization bound for unforeseen threat models with the knowledge of a source model. The paper further provides a training method for achieving better unforeseen robustness using the generalization bound.

**Questions:**

Questions about theorems:

1. I don't fully understand the $\hat{L}_S(f)$. Is it the AT-VR loss in the paper?

2. How can theorem 4.2 guarantee that both \pho and \sigma_G are small enough to make the first term <<1?

3. The paper does not mention too much about the sample generalization gap, and the results are all on the threat model generalization gap. Is it possible that the threat model generalization gap is decreasing while the sample generalization gap is increasing?

4. The existence of the expansion function guarantee seems very weak.

Questions about experiments:

1. AT-VR seems much more computationally expensive than AT.

2. The paper needs to compare its method with state-of-art robust training. Only comparing to AT (2017) is not enough.

3. The experimental results are weak. For example, the clean acc drops 10% compared to AT. Note that AT itself has a significant clean acc drop. This trade-off is not acceptable.

4. Why VR is applied on logits instead of the input of the fully-connected layers?

**Limitations:**

Yes

**Strengths And Weaknesses:**

Strengths: Achieving a generalization bound to unforeseen attacks is challenging and interesting. The paper provides both theoretical results and experimental results.

weaknesses: The theorems are not clear; The experimental results are weak; The comparisons are insufficient.

---

> ### Author Response · Authors · 2022-08-02
> **Author response (pt 1)**
>
> Thank you for the feedback and questions.  We are pleased that you find this direction of research interesting.  We provide our responses to your comments and questions below:
>
> > I don't fully understand the $\hat{L}_S(f)$. Is it the AT-VR loss in the paper?
>
> The $\hat{L}_S(f)$ term represents the empirical adversarial risk measured across the source threat model:
>
> $\hat{L}_S(f) = \frac{1}{n} \sum_{i=1}^n \max_{\hat{x} \in S(x_i)} \ell(f(\hat{x}), y_i )$
>
> This is similar to the standard AT loss which approximates the $\hat{x}$ with PGD.
> The AT-VR loss is given in the equation at the top of Section 5.1 and can be written as
>
> $\hat{L}_S(f) + \frac{\lambda}{n} \sum_{i=1}^n \max_{x_1, x_2 \in S(x_i)} ||h(x_1) - h(x_2)||_2$
>
> > How can theorem 4.2 guarantee that both \pho and \sigma_G are small enough to make the first term <<1?
>
> Theorem 4.2 makes no guarantees that $\rho\sigma_G\epsilon(T, m) << 1$.  Rather, Theorem 4.2 makes a statement about the connection between variation and threat model generalizability.  Here $\epsilon(T, m)$ is a function that upper bounds $V(h, T)$ with probability $1-\delta$.  For example, we can take $\delta=0$ and $\epsilon(T, m) = \max_{h \in \mathcal{H}} V(h, T)$.  The definition of $\epsilon(T, m)$ is what motivates the use of variation regularization: if we can encourage the learning algorithm to only choose functions with small variation, then we can have a smaller upper bound ($\epsilon(T, m)$) on $V(h, T)$, which gives better generalizability since we reduce the value of $\rho\sigma_G\epsilon(T, m)$.
>
> > Is it possible that the threat model generalization gap is decreasing while the sample generalization gap is increasing?
>
> We do not discuss the sample generalization gap in much depth since this is a quantity that has been studied in prior works [Attias et al. 2019, Schmidt et al. 2018].  This quantity represents how close the empirical adversarial risk $\hat{L}_S(f)$ is to the expected adversarial risk $L_S(f)$ and can generally be reduced by either 1) reducing the complexity of the hypothesis class $\mathcal{F}$ or 2) increasing number of samples used.  Meanwhile, the threat model generalization gap depends mainly on the choice of source and target threat model and the quality of the model returned by the learning algorithm.  Since VR is a regularization-based approach, our algorithm reduces the complexity of $\mathcal{F}$ and thus should also improve the sample generalization gap.
>
> Attias, Idan, Aryeh Kontorovich, and Yishay Mansour. "Improved generalization bounds for robust learning." Algorithmic Learning Theory. PMLR, 2019.
> Schmidt, Ludwig, et al. "Adversarially robust generalization requires more data." Advances in neural information processing systems 31 (2018).
>
> > The existence of the expansion function guarantee seems very weak.
>
> While Theorem 4.7 only demonstrates that an expansion function exists for linear models, we provide experimental results in Figure 2 to show that for ResNet-18 a linear expansion function also exists.  In our response to Reviewer PVxo, we also provide additional values for slopes of the expansion function between $\ell_p$ and StAdv threat models to further demonstrate that even between these threat models an expansion function with small slope exists.
>
> > AT-VR seems much more computationally expensive than AT.
>
> We agree that AT-VR is more computationally expensive compared to regular adversarial training.  As per reviewer feedback, we have added a discussion of this into Section 6.
> Interestingly, we note that in some special cases, AT-VR may be less expensive than regular adversarial training since fewer iterations of PGD are required.  For example, when training with an $\ell_2$ source, we find that only a single iteration of PGD is needed for AT-VR:
>
> | lambda | PGD iterations | Clean Acc | Source Acc | Linf target acc | L2 target acc | StAdv target acc | Recolor target acc | Union acc |
> |--------|----------------|-----------|------------|-----------------|---------------|------------------|--------------------|-----------|
> | 0      | 1              | 89.00     | 66.53      | 5.54            | 31.55         | 0.26             | 33.43              | 0.05      |
> | 0      | 3              | 88.72     | 67.58      | 7.07            | 35.47         | 0.55             | 36.41              | 0.18      |
> | 0      | 10             | 88.49     | 66.65      | 6.44            | 34.72         | 0.76             | 66.52              | 0.33      |
> | 1      | 1              | 86.88     | 67.00      | 11.52           | 37.24         | 38.34            | 64.44              | 10.09     |
>
> We would also like to comment that in some safety-critical applications where robustness is of high importance (ie. autonomous vehicles) this computational expense can be worthwhile.

---

> > ### Author Response · Authors · 2022-08-02
> > **Author response (pt 2)**
> >
> > > The paper needs to compare its method with state-of-art robust training. Only comparing to AT (2017) is not enough.
> >
> > While Table 1 in the main text contains comparison to on AT, we include additional comparisons to other training techniques such as TRADES (Appendix F.1.7) [Zhang et al. 2019]  and perceptual adversarial training (PAT) [Laidlaw et al. 2021], the state-of-the-art for robustness against unforeseen attacks, in Appendix F.1.8.  Additionally, we note that variation regularization can be applied on top of any robust training technique, which is why we compare AT-VR to AT, TRADES-VR to TRADES, and PAT-VR to PAT.
> >
> > Based on reviewer feedback, we will be moving the PAT-VR results from the appendix into the main text in the camera ready version since we agree that this is an important comparison and we are allowed an additional content page in the camera ready paper.
> >
> > Zhang, Hongyang, et al. "Theoretically principled trade-off between robustness and accuracy." International conference on machine learning. PMLR, 2019.
> > Laidlaw, Cassidy, Sahil Singla, and Soheil Feizi. "Perceptual Adversarial Robustness: Defense Against Unseen Threat Models." International Conference on Learning Representations. 2020.
> >
> > > The experimental results are weak. For example, the clean acc drops 10% compared to AT. Note that AT itself has a significant clean acc drop. This trade-off is not acceptable.
> >
> > We respectfully disagree.  In Table 1, we find that AT-VR significantly increases robust accuracy on unforeseen threat models over adversarial training (this increase is as much as 33% for generalizing to the StAdv threat model).  In Table 2 in the Appendix, we also show error bars for ResNet-18 CIFAR-10 models to further demonstrate that the increase in robustness from variation regularization is significant.
> > In regards to the drop in clean accuracy, we note that regularization strength $\lambda$ is a tunable hyperparameter, and as shown in Table 3 of the Appendix, using smaller values of $\lambda$ results in smaller drops in clean accuracy (but smaller improvement in unforeseen robustness).  We also note that other methods for robustness against unforeseen attacks, such as PAT, also exhibit a tradeoff with clean accuracy.  For example, a ResNet-50 trained using PAT-AlexNet achieves 71.6% clean accuracy while training ResNet-50 with $\ell_{\infty} or $\ell_2$ adversarial training gives ~85% clean accuracy.  In fact, we find that PAT-VR (Table 8 in Appendix) does not trade off additional clean accuracy (PAT-VR achieves 72.5% clean accuracy) in comparison to PAT.
> > Additionally, we argue that whether this tradeoff is acceptable or not depends on application.  In settings where safety is a primary concern, we may be willing to trade off accuracy to improve robustness.
> >
> > > Why VR is applied on logits instead of the input of the fully-connected layers?
> >
> > From Theorem 4.2, we have the result that for any model that can be composed via a feature extractor and top level classifier, learning algorithm A $(\rho\sigma_G\epsilon(T, m), \delta)$-robustly generalizes from source threat model S to target threat model T.  Since we want to reduce the generalization gap, we would like the $\rho\sigma_G\epsilon(T, m)$ to be small.  We decrease $\epsilon(T, m)$ via variation regularization, $\rho$ comes from the chosen loss function (cross entropy loss), so the last term we would like to be small is $\sigma_G$.  We approach this by fixing the top level classifier to just be an identity function, which sets $\sigma_G$ to be 1.
> > We do include results with VR applied to the input of fully-connected layers (see Appendix F.2) and observe improvements in unforeseen robustness over AT in that setting as well.

---

> > > ### Comment · Reviewer_UuTG · 2022-08-09
> > > **Follow-up questions**
> > >
> > > I want to thank the authors for the detailed response. However, I still have a few concerns. I checked Table 3 of the Appendix and found that the clean accuracy under (CIFAR-10, ResNet-18, $\ell_\infty$, $\lambda=1.0$) is 79.72. But Table 1 gives 72.91 when $\lambda=0.5$. This seems confusing and does not satisfy "using smaller values of $\lambda$ results in smaller drops in clean accuracy." I also noticed that the authors listed some papers related to the Defense of Multiple Perturbations. Has the proposed method been compared with these Multiple Perturbation Defense strategies?

---

> > > > ### Author Response · Authors · 2022-08-09
> > > > **Response to follow-up questions**
> > > >
> > > > Thank you for the follow up questions.  Please see our responses below:
> > > >
> > > > 1. Clean accuracy trend: In Appendix Table 3, there is no entry for (CIFAR-10, ResNet-18, $\ell_{\infty}$, $\lambda=1.0$).  The highest value provided is for $\lambda=0.5$ and the trend of smaller values of lambda leading to smaller drops in clean accuracy is consistent.  Are you referring to Appendix Table 9 instead?  In Appendix Table 9, variation regularization is performed on the features before the final fully connected layer of the network architecture, while for Table 1 and Table 3, variation regularization is performed on the logits.  Because regularization is applied at different levels between these tables, it is difficult to directly compare $\lambda = 0.5$ in Table 1 to $\lambda=1$ in Table 9.  However, we note that for when comparing multiple $\lambda$ values within Table 9, we also observe that using smaller values of lambda leads to smaller drops in clean accuracy.
> > > >
> > > > 2. Comparing to defenses against multiple perturbations: To clarify, our work differs from multiple perturbation defense strategies since we do not use knowledge of the target threat models used during evaluation, while multiple perturbation defenses assume that we have full knowledge of those threat models and train directly with access to them.  In our response to Reviewer 9qwD, we we provide accuracies for ResNet-18 model trained on CIFAR-10 with access to all target threat models in Table 1 (training using Tramer et al. 2019 MAX technique).  We copy the results here:
> > > >
> > > > | Clean Acc | Linf ($\epsilon=\frac{12}{255}$) Acc | L2 ($\epsilon=1$) Acc | StAdv Acc | Recolor Acc | Union Acc |
> > > > |-----------|--------------------------------------|-----------------------|-----------|-------------|-----------|
> > > > | 77.81     | 41.76                                | 36.74                 | 38.90     | 63.99       | 27.23     |
> > > >
> > > > We note that the union accuracies from using multiperturbation robustness techniques should be viewed as an upper bound on performance gain.  We are also working on obtaining corresponding accuracies for the other architectures listed in Table 1 into the Appendix for the camera-ready paper.

---

> > > > > ### Comment · Reviewer_UuTG · 2022-08-09
> > > > > **Increasing my score**
> > > > >
> > > > > Thanks for these answers. I will increase my score from 4 to 5.

---

> ### Author Response · Authors · 2022-08-08
> **Any comments or additional questions?**
>
> Dear reviewer UuTG,
>
> Given that the author-reviewer discussion period is ending soon, we wanted to see if our responses in our rebuttal address all your questions and concerns about our paper.  If you have any additional comments or questions, we are happy to address those as well in the remaining time.
>
> Thank you

---

### Official Review · Reviewer_PVxo · 2022-07-11

**Rating:** 7
**Confidence:** 5
**Soundness:** 3 good
**Presentation:** 3 good
**Contribution:** 4 excellent

**Summary:**

This paper seeks to theoretically formulate the problem of learning a classifier which is robust against unseen attacks—that is, a threat model which is not used at training time. The authors define a threat model as a set of allowed perturbations to an input, and distinguish a source threat model used for training versus a target threat model used for evaluation. They characterize the target threat model as "unseen" if it is a strict superset of the source threat model for all inputs. Then, the generalization gap between robust error on the training set using the source threat model and robust error on a test set using the target threat model can be decomposed into a standard sample generalization term and a new threat model generalization term. The authors show that the second term can be bounded if there is an "expansion function" which controls how much more "variation" the classifier exhibits on the target threat model as compared to the source threat model, and give examples of when such an expansion function exists or does not exist. Finally, the authors propose a method AT-VR which explicitly regularizes the variation on the source threat model, which according to their theory should improve generalization to unseen threat models. They show that it does empirically improve unseen threat model accuracy across various attacks and datasets.

**Questions:**

 * Is there any empirical explanation you can give for why the expansion function in practice is linear with low slope, even though your theory doesn't apply directly to neural networks? For instance, suppose you take the linear approximation of a trained network around each test point and apply Theorem 4.7 to get an approximate local expansion function based on the linear approximation, then take the average over the test points. Does this give you something similar to what you observe empirically in, e.g., Figure 2?
 * Can you compute the expansion function slopes for non $\ell_p$ threat models? For instance, it would be interesting to see what the expansion function looks like for $\ell_\infty$ to StAdv or vice versa. This would also potentially be a more compelling case for expansion functions since $\ell_p$ threat models are somewhat similar already (since one can bound any $\ell_p$ norm by a multiple of an $\ell_q$ norm). Instead, looking at two very different threat models like $\ell_\infty$ and StAdv or $\ell_2$ and StAdv would give more evidence that expansion functions exist between many threat models. Another interesting example would be measuring the expansion function between LPIPS-based attacks and $\ell_p$ or StAdv attacks.
 * What happens if you plug in the expansion function slope calculated empirically into Corollary 4.4? Does this give a reasonable estimate for the adversarial error on the target threat model? If not, what do you think accounts for the gap?

Small thing: on line 158, I think you can only conclude that $L_T(f) \geq L_S(f)$, not that it is strictly greater. For instance if $f$ outputs the constant uniform distribution over classes for every input, then $L_T(f) = L_S(f)$ for any $S, T$.

**Limitations:**

I think there are a couple of limitations that the authors did not discuss. While these limitations don't detract too much from the paper's significance, it would be good to mention and address them:
 * The method AT-VR seems to be much more expensive than regular adversarial training (it should take 3x the computation if I'm understanding correctly since one needs to compute gradients wrt three inputs instead of one for each attack step). I don't think the authors mention this.
 * As I mentioned above, the theoretical bounds on the expansion function are not directly applicable to neural networks. Furthermore, it's unclear if the empirical expansion functions predict empirical performance using the bounds in the paper.

**Strengths And Weaknesses:**

Strengths:
 * The problem addressed—adversarial robustness against unseen threat models—is both difficult and understudied compared to other areas of adversarial robustness. The formulation proposed in this paper seems helpful to concretize the problem and provide directions for further research.
 * I particularly appreciated the formulation of the expansion function, which is empirically measurable and can help practitioners working on this problem to diagnose why their model may not be generalizing across threat models.
 * The empirical results seem strong—the proposed method AT-VR generally helps across many threat models and rarely hurts. I also appreciate the very extensive experiments with other threat models and alternative training methods in the appendix.

Weaknesses:
 * While the theoretical bounds using variation and expansion functions give some intuition, they are probably vacuous for reasonable learning problems. Furthermore, none of the bounds on the expansion function apply to neural networks (at least to my understanding, correct me if I'm wrong).

---

> ### Author Response · Authors · 2022-08-02
> **Author response (pt 1)**
>
> Thank you for the feedback and questions. We are encouraged that you find this direction of research interesting, appreciate the expansion function formulation, and find our experimental results strong.  We also greatly appreciate the suggestions for additional experiments for investigating the connection between our theoretical results and experimental results. We provide our responses to your questions and suggestions below:
>
> > Can you compute the expansion function slopes for non-lp threat models?
>
> We report the computed expansion function slopes for lp to StAdv threat model below:
> | Source     | Target     | Expansion function slope |
> |------------|------------|--------------------------|
> | Linf 8/255 | StAdv 0.05 | 1.29                     |
> | L2 0.5     | StAdv 0.05 | 14.32                    |
>
> For these source and target pairs, we find that the linear expansion function is a good fit.  The corresponding plots have also been added to the paper Appendix G.1 of the uploaded revision.
>
> We also compute expansion function slopes for StAdv to other threat models.  We note that for computing the expansion function, we take the models saved every 5 epochs for generating the results for StAdv in Table 7.  This gives 60 points for estimating the expansion function.  We report results below:
>
> | Source     | Target     | Expansion function slope |
> |------------|------------|--------------------------|
> | StAdv 0.03 | Linf 8/255 | 28.61                    |
> | StAdv 0.03 | L2 0.5     | 3.64                     |
> | StAdv 0.03 | StAdv 0.05 | 2.80                     |
>
> We find that when the source is StAdv, a linear expansion function does not fit the trend between source and target variation well.  A better model would be a log function since the slopes at points where source variation is closer to 0 is much larger than the slopes computed at points further from 0.  The corresponding plots have been added to Appendix G.1 of the uploaded revision.
>
> > What happens if you plug in the expansion function slope calculated empirically into Corollary 4.4? Does this give a reasonable estimate for the adversarial error on the target threat model? If not, what do you think accounts for the gap?
>
> We present the predicted losses below.  Here, we use the expansion functions computed for ResNet-18 models to predict the cross entropy loss for ResNet-18 CIFAR-10 models in Table 1:
>
> For Linf source with $\epsilon=8/255$:
> | Target      | Model source variation | Model source loss | Predicted target loss (Corollary 4.4) | True target loss | Gap   |
> |-------------|------------------------|-------------------|---------------------------------------|------------------|-------|
> | Linf 16/255 | 4.90                   | 0.93              | 12.85                                 | 2.44             | 10.41 |
> | L2 0.5      | 4.90                   | 0.93              | 7.86                                  | 0.93             | 6.93  |
> | StAdv       | 4.90                   | 0.93              | 9.87                                  | 5.13             | 4.74  |
> | Linf 16/255 | 0.98                   | 1.26              | 3.64                                  | 1.76             | 1.88  |
> | L2 0.5      | 0.98                   | 1.26              | 2.64                                  | 1.27             | 1.37  |
> | StAdv       | 0.98                   | 1.26              | 3.05                                  | 2.11             | 0.94  |
>
> For L2 source with $\epsilon = 0.5$:
> In general, we find that for models with smaller variation (trained with variation regularization), the loss estimate using the slope from the expansion function generally improves.  In the case where the target threat model is Linf, we believe that the large gap between predicted and true loss for the unregularized model stems from the fact that we model the expansion function with a linear model.  From Figure 2, we can see that at larger values of source variation the linear model for expansion function becomes an increasingly loose upper-bound.  Improving the model for expansion function (ie. using a log function) may reduce this gap.
>
> We have added these results into Appendix G.2 (we will reorganize the Appendix in the camera ready submission).

---

> > ### Author Response · Authors · 2022-08-02
> > **Author response (pt 2)**
> >
> > > Limitations: computational complexity, theoretical bounds not applicable to NN, may be loose
> >
> > Thank you for pointing out these additional limitations.  We have updated Section 6 to discuss these as well.  The results in our response to the previous question should also further address the question of how predictive the empirical expansion functions are.
> >
> > On the topic of computational complexity of AT-VR, since we use the same number of PGD iterations for finding the adversarial example as computing VR, it takes 3x as long.  Interestingly, we note that in some cases we can reduce the number of iterations of PGD used and achieve similar gains in unforeseen robustness.  For example, when training with an L2 source, we find that only a single iteration of PGD is needed for AT-VR, which has about the same computational complexity as 3-iteration PGD.
> >
> > | lambda | PGD iterations | Clean Acc | Source Acc | Linf target acc | L2 target acc | StAdv target acc | Recolor target acc | Union acc |
> > |--------|----------------|-----------|------------|-----------------|---------------|------------------|--------------------|-----------|
> > | 0      | 1              | 89.00     | 66.53      | 5.54            | 31.55         | 0.26             | 33.43              | 0.05      |
> > | 0      | 3              | 88.72     | 67.58      | 7.07            | 35.47         | 0.55             | 36.41              | 0.18      |
> > | 0      | 10             | 88.49     | 66.65      | 6.44            | 34.72         | 0.76             | 66.52              | 0.33      |
> > | 1      | 1              | 86.88     | 67.00      | 11.52           | 37.24         | 38.34            | 64.44              | 10.09     |
> >
> > We have incorporated these results into Appendix G.3.
> >
> > > Small thing: on line 158, I think you can only conclude that L_T(f) \ge L_S(f), not that it is strictly greater.
> >
> > Thank you for pointing this out.  We have updated line 158 to state that $L_T(f) \ge L_S(f)$.

---

> > > ### Author Response · Authors · 2022-08-02
> > > **Author response (pt 3)**
> > >
> > > > Is there any empirical explanation you can give for why the expansion function in practice is linear with low slope, even though your theory doesn't apply directly to neural networks? For instance, suppose you take the linear approximation of a trained network around each test point and apply Theorem 4.7 to get an approximate local expansion function based on the linear approximation, then take the average over the test points. Does this give you something similar to what you observe empirically in, e.g., Figure 2?
> > >
> > > While Theorem 4.7 states that a linear expansion function exists for linear feature extractors, the slope of the expansion function derived in the proof  (Lemma D.2 and D.4 in Appendix) may be larger than the smallest linear expansion function which was plotted empirically.  For example, if we consider a toy example with a linear feature extractor $h: \mathbb{R}^{25} \to \mathbb{R}^2$ for a task of classifying data between 2 Gaussians (setup described in Appendix E.1), we obtain the following results:
> > >
> > > | Source (epsilon) | Target (epsilon) | Predicted expansion function slope | Minimum (empirical) expansion function slope | Ratio (predicted/empirical) |
> > > |------------------|------------------|------------------------------------|----------------------------------------------|-----------------------------|
> > > | L2 (0.01)        | L2 (0.05)        | 8.61                               | 5.02                                         | 1.72                        |
> > > | Linf (0.01)      | Linf (0.05)      | 54.77                              | 4.88                                         | 11.22                       |
> > > | L2 (0.01)        | Linf (0.05)      | 44.84                              | 20.72                                        | 2.16                        |
> > > | Linf (0.01)      | L2 (0.05)        | 10.18                              | 1.39                                         | 7.32                        |
> > >
> > > This gap arises from the fact that we make no assumptions on the data distribution.  For Linf to Linf, L2 to Linf, and Linf to L2, the computed expansion function slope scales with the square root of the dimension of the input, so the predicted slope becomes meaningless for high dimensional data such as CIFAR-10.  However, we provide results based on your suggestion of linear approximation for L2 source to L2 target for which the predicted slope is independent of input dimension.  Specifically, we do the following:
> > >
> > > For each ResNet-18 model (used for plotting empirical expansion function):
> > >
> > > - For each test point, take the local approximation of the model around that test point and compute the condition number of this approximation
> > > - Average all computed local condition numbers to obtain a condition number for the model
> > > - Take the maximum out of all model condition numbers to approximate $B$, the upper bound on condition number for the hypothesis class
> > > - Estimate expansion function slope via $B\frac{\epsilon_1}{\epsilon_2}$
> > >
> > > Doing this, we find that the estimated $B$ ends up being very large (B=224334), so the predicted expansion slope also ends up being much larger than found empirically (448668 vs 1.14).  Thus with the current theoretical results, we do not have a good explanation for why we observe linear expansion functions for neural networks.  This however would make an interesting direction for future research.

---

> ### Comment · Reviewer_PVxo · 2022-08-06
> **Response to authors and overall review for AC**
>
> Thanks to the authors for your responses to my comments and questions. I am mostly satisfied with the authors' responses; I appreciate the experiments with StAdv (I actually think the StAdv -> Lp expansion function looks pretty linear to me despite the larger slope around 0). I still have some concerns that the slope of the expansion function cannot always explain in practice which source threat model generalize to which target ones (e.g., the theory predicts L2 robustness will transfer better to StAdv than Linf, even though the opposite is true in practice). However, overall, I think the idea introduced is valuable and the variation regularization technique seems to work quite well. I vote for acceptance and hope this paper will spur more investigations into why these expansion functions exist and the theory of adversarial robustness against unseen attacks in general.

---

### Meta-Review · Area_Chair_GBCX · 2022-08-23

**Recommendation:** Accept
**Confidence:** Certain

**Metareview:**

This paper seeks to theoretically formulate the problem of learning a classifier which is robust against unseen attacks. Reviewers generally liked the way authors formulated this understudied problem in adv robustness and found the paper well written. There were some questions and concerns regarding the usefulness of these results in reasonable learning problems as well as its numerical analysis. Given all, I think the paper is above the accept threshold.

**Award:**

No

---

### Decision · Program_Chairs · 2022-09-14

Accept